# S3OD: Towards Generalizable Salient Object Detection with Synthetic Data

**Orest Kupyn[1], Hirokatsu Kataoka[1, 2], Christian Rupprecht[1]**
[1]University of Oxford, VGG [2]AIST
`https://s3odproject.github.io`

## Abstract

Salient object detection exemplifies data-bounded tasks where expensive pixel-precise annotations force separate model training for related subtasks like DIS and HR-SOD. We present a method that dramatically improves generalization through large-scale synthetic data generation and ambiguity-aware architecture. We introduce S3OD, a dataset of over 139,000 high-resolution images created through our multi-modal diffusion pipeline that extracts labels from diffusion and DINO-v3 features. The iterative generation framework prioritizes challenging categories based on model performance. We propose a streamlined multi-mask decoder that handles the inherent ambiguity in salient object detection by predicting multiple valid interpretations. Models trained only on synthetic data achieve 20-50% error reduction in cross-dataset generalization, while fine-tuned versions reach state-of-the-art performance across DIS and HR-SOD benchmarks.

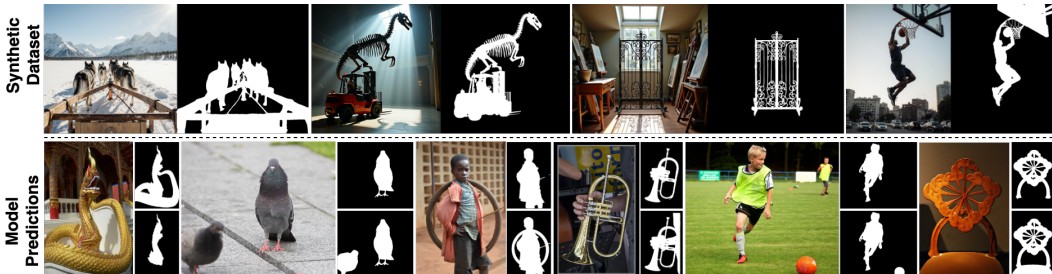

Figure 1: *S3OD* Top: Our large-scale synthetic dataset, consisting of diverse complex scenes and high-quality samples. Bottom: Model Predictions. Our model trained on synthetic data generalizes well to real-world images, handling ambiguous scenes by predicting alternative hypotheses.

## 1 Introduction

Salient object detection (SOD) is a fundamental problem in computer vision with applications spanning AR/VR (Tian et al., 2022), robotics (Chan & Riek, 2020), 3D reconstruction (Liu et al., 2021a), and image editing (Goferman et al., 2011). Recently, two specialized subtasks have emerged: dichotomous image segmentation (DIS), which focuses on highly accurate boundaries, and high-resolution SOD (HR-SOD) for 2K-8K resolution images, both of which present new generalization challenges. SOD exemplifies tasks that are fundamentally limited by the availability of labeled data. Creating diverse, representative datasets is challenging, requiring extensive coverage of real-world scenarios and object types. The labeling process requires pixel-precise manual annotations, which can take up to 10 hours per sample (Qin et al., 2022). Moreover, annotations often exhibit inherent ambiguities and inconsistencies across datasets, as annotators interpret scene saliency differently, posing a fundamental challenge for deterministic approaches. These constraints yield relatively small datasets (Qin et al., 2022; Zeng et al., 2019) that cannot capture the complexity of real-world systems. Even large-scale datasets like SA-1B (Ravi et al., 2024) struggle with the high-resolution pixel-perfect data (Ke et al., 2023). Current approaches train separate models for DIS and HR-SOD due to small datasets and domain gaps, leading to task-specific overfitting rather than generalizable principles. Recent architectural innovations (Yu et al., 2024; Zheng et al., 2024; Kim et al., 2022) have achieved incremental improvements but have not addressed cross-domain generalization. The

fundamental bottleneck remains data scarcity, not model complexity, whereas models typically enforce deterministic predictions, thereby ignoring ambiguity. Synthetic data offers an attractive solution, but existing approaches have critical limitations. Traditional pseudo-labeling setups are limited by the teacher's capabilities and often rely on the same vision encoders, thereby imposing performance ceilings. Methods that extend diffusion models to predict masks directly (Wu et al., 2023a) suffer from consistency issues arising from noisy diffusion features. In contrast, mask-conditioned generation (Qian et al., 2024) struggles with diversity as obtaining large mask libraries and generating complex scenes remain challenging.

In this work, we aim to unify DIS and HR-SOD by addressing two main limitations of prior work. We refer to the unified task as high-fidelity salient segmentation. To this end, we introduce: 1) a multi-modal data generation pipeline that leverages the generative power of diffusion models, eliminating teacher bottlenecks, 2) an ambiguity-aware architecture handling multiple interpretations, and 3) an iterative generation framework adapting to model weaknesses. Our main contributions are:

**Multi-Modal Dataset Diffusion Pipeline:** Our diffusion pipeline simultaneously generates images and masks by extracting FLUX DiT feature maps, concept attention maps, and DINO-v3 (Siméoni et al., 2025) representations during the generation process. The generation pipeline leverages rich spatial understanding encoded during generation, along with robust semantic features from discriminative models, to decode high-quality masks. This ensures strong image-label alignment, enabling a flexible framework applicable to other dense prediction tasks.

**Iterative Generation Framework:** We introduce feedback-driven synthetic data generation that dynamically identifies model weaknesses, continuously adapting sampling distribution to prioritize challenging categories. Unlike traditional static methods, this iterative approach enables continuous improvement as datasets grow.

**Large-Scale Synthetic Dataset:** Using our pipeline, we generate 139,000+ high-resolution images with pixel-wise annotations, over $2\times$ more than all existing SOD datasets combined. This enables up to a 50% reduction in error across benchmarks when evaluated for cross-dataset generalization. Models trained solely on synthetic data achieve strong cross-dataset generalization without real training data, whereas fine-tuned versions reach state-of-the-art performance on the DIS and HR-SOD benchmarks.

**Ambiguity-Aware Architecture:** We directly address SOD's inherent ambiguity through a multi-mask decoder allowing multiple valid interpretations while enabling a simpler architecture compared to current state-of-the-art methods. We employ the DINO-v3 backbone, which leverages enhanced visual representations to improve generalization.

## 2 RELATED WORK

**Salient Object Detection:** SOD has evolved from handcrafted features (Borji et al., 2015) to complex multi-view transformer architectures (Yu et al., 2024). BASNet (Qin et al., 2019) introduced boundary-aware refinement with hybrid loss functions for precise object segmentation, while subsequent work (Zhao et al., 2019; Wei et al., 2020b; Wu et al., 2019b; Feng et al., 2019) explored efficient edge-refinement strategies. $U^2$-Net (Qin et al., 2020) developed a nested UNet architecture to capture multi-scale contextual information. CPD (Wu et al., 2019a) introduced cascaded decoders that directly refine features using generated saliency maps. PFANet (Zhang et al., 2018) and PAGENet (Wang et al., 2019) leveraged pyramid attention networks to enhance segmentation quality. However, these approaches remain constrained by limitations of the training dataset and struggle in high-resolution inference scenarios. Recently, HR-SOD and DIS emerged as specialized subtasks focused on high-resolution, accurate segmentation. IS-Net (Qin et al., 2022) established the DIS baseline using intermediate supervision with feature-level and mask-level guidance. Newer approaches incorporated transformer backbones (Liu et al., 2021b) to enhance feature extraction. InSPyReNet (Kim et al., 2022) adopted an image pyramid architecture for HR-SOD, whereas BiRefNet (Zheng et al., 2024) introduced bilateral reference frameworks to capture intricate details. MVANet (Yu et al., 2024) recently proposed multi-view aggregation to detect finer details while improving efficiency. Nevertheless, these methods produce a single deterministic output and are constrained by limited training data. Our approach addresses both limitations while simplifying the architecture.

**Synthetic Data Generation:** Diffusion models have transformed data generation by enabling high-quality, diverse synthetic datasets. Recent work (Shipard et al., 2023; Sarıyıldız et al., 2023; Tian

et al., 2023; Azizi et al., 2023; Fan et al., 2024) improved classification model performance by generating synthetic data with latent diffusion models (Rombach et al., 2022), though this approach is limited to image classification. DiffuMask (Wu et al., 2023b), Attn2mask (Yoshihashi et al., 2024), and DatasetDM (Wu et al., 2023a) utilize diffusion models to generate synthetic images with annotations for segmentation tasks. However, DatasetDM's attention-based extraction yields noisy, incomplete masks with imprecise boundaries and struggles with complex multi-object scenes. OVDiff (Karazija et al., 2024) synthesises support image sets for arbitrary textual categories, while Instance Augmentation (Kupyn & Rupprecht, 2024) provides augmentation frameworks but only slightly expands original distributions. VGGHeads (Kupyn et al., 2024) demonstrated the impact of synthetic data on generalization for 3D head modeling, but remains bounded by external teacher models. For SOD specifically, SODGAN (Wu et al., 2022) employs GANs but struggles with complex scenes due to limited variability in training data. MaskFactory (Qian et al., 2024) conditions image generation on edited masks, but is limited to producing only slight variations of the training set. Unlike approaches that rely on noisy attention extraction, mask conditioning, or external teacher models, our method extracts supervision from multiple complementary sources within the generative process itself. By combining DINO-v3 visual features (Siméoni et al., 2025), diffusion transformer activations, and concept attention maps (Helbling et al., 2025), we obtain robust supervision with strong image-mask alignment while eliminating performance bottlenecks.

## 3 MODEL

Most recent SOD methods focus on improving performance by incorporating complex architectural components, such as multi-view feature fusion (Yu et al., 2024) or iterative refinement modules (Zheng et al., 2024). In contrast, we propose a lightweight architecture that addresses SOD ambiguity through a multi-mask decoder while significantly simplifying other components.

### 3.1 MODEL ARCHITECTURE

We build our model upon the Dense Prediction Transformer (DPT) (Ranftl et al., 2021) architecture, which processes input images through transformer (Vaswani et al., 2017) stages followed by multi-scale feature reassembly. DPT transforms the input into patch-token sequences, processes them through transformer layers, and then reshapes them into multi-scale image-like representations. These features are progressively fused and upsampled through residual convolutional blocks (He et al., 2016) to produce final predictions. We adopt this efficient hierarchical design as our backbone. We initialize the DPT encoder with DINO-v3 weights to improve generalization, leveraging visual representations from large-scale self-supervised training. The full architecture is shown in Figure 2.

We formulate the problem as function $f : \mathcal{I} \rightarrow \mathcal{M}$ mapping from images $\mathcal{I} \subset \mathbb{R}^{H \times W \times 3}$ to binary masks $\mathcal{M} = \{0, 1\}^{H \times W}$ of spatial resolution $H \times W$. Many training annotations are ambiguous: multiple objects may be present, with unclear saliency interpretations. Single-output models often average all possible predictions, leading to low-confidence regions.

To address this, we design the final mask prediction head to output multiple masks $(m_1, \ldots, m_N)$. Predicted masks are soft $m_i \in (0, 1)^{H \times W}$ to model pixel-wise confidence. For each training image $I \in \mathcal{I}$, only one ground truth annotation $y \in \mathcal{M}$ is available. Inspired by multiple-choice learning (Guzman-Rivera et al., 2012), during training, the main loss applies to the best prediction $i^* = \arg\min_i \text{IoU}(m_i, y)$, chosen via IoU score between predicted and ground truth masks.

To prevent unused branches from degrading, we employ relaxed assignment (Rupprecht et al., 2017) where loss is computed across all branches with decaying weight: $\mathcal{L} = \mathcal{L}_{i^*} + \lambda e^{-\gamma t} \sum_i^N \mathcal{L}_i$, where $\lambda$ controls initial auxiliary branch weight, $\gamma$ is decay rate, $t$ is current epoch. Individual losses $\mathcal{L}_i = \mathcal{L}(m_i, y)$ are described next. For test-time selection, the model estimates IoU scores $(s_1, \ldots, s_N)$ for every prediction. This is supervised by actual IoU scores between prediction and ground truth during training, and this estimate is used to select the highest-scoring mask during testing.

### 3.2 OBJECTIVE FUNCTION

Following standard semantic segmentation practice, we employ a multi-component loss combining pixel-wise and region-wise supervision. The total loss $\mathcal{L}$ consists of two main components: **Focal Loss** (Lin et al., 2017) $\mathcal{L}_{\text{focal}}$ for class imbalance and **IoU Loss** $\mathcal{L}_{\text{IoU}}$ for region-level accuracy.

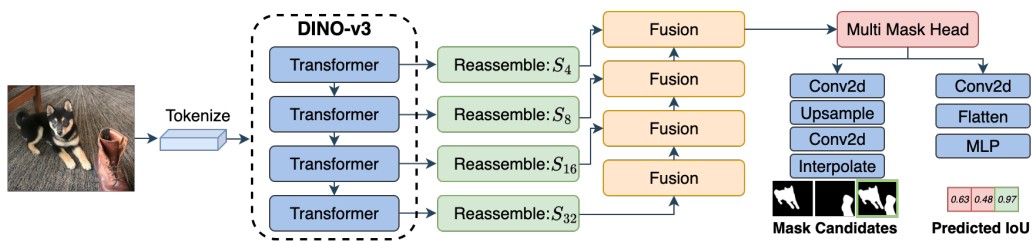

Figure 2: **S3ODNet Architecture.** Model extends DPT (Ranftl et al., 2021) to predict multiple mask candidates and a vector of IoUs with the ground truth, employing DINO-v3 as the backbone. During training, the loss is propagated through the branch with the highest predicted IoU.

**Focal Loss.** To address foreground-background imbalance, we implement focal loss, widely used in dense prediction:

$$\mathcal{L}_{\text{focal}}(m_i) = - \sum_{p=1}^{H \times W} (1 - m_i(p))^\tau y(p) \log(m_i(p))$$

where $p$ iterates over pixels indexing predicted mask $m_i(p)$ and ground truth $y(p)$, and $\tau = 2$ is the focusing parameter.

**IoU Loss.** To capture region-level accuracy, we incorporate IoU loss, measuring overlap between predicted and ground truth masks:

$$\mathcal{L}_{\text{IoU}}(m_i) = 1 - \frac{\sum_{p=1}^{H \times W} m_i(p) y(p)}{\sum_{p=1}^{H \times W} (m_i(p) + y(p) - m_i(p)y(p))}$$

The overall mask loss combines both components:

$$\mathcal{L}_{\text{mask}}(m_i) = \lambda_{\text{mask}} \mathcal{L}_{\text{focal}}(m_i) + \mathcal{L}_{\text{IoU}}(m_i, y)$$

where $\lambda_{\text{mask}} = 10$ balances the losses.

**IoU Score Loss.** To enable optimal mask selection at inference, we supervise predicted IoU scores $s_i$ using mean squared error between predicted and actual IoU values:

$$\mathcal{L}_{\text{score}}(s_i) = (s_i - \text{IoU}(m_i, y))^2$$

Finally, the overall training objective comprises the mask loss of best prediction, score loss for all predictions, and a decaying regularizer across all predicted masks:

$$\mathcal{L}_{\text{mask}}(m_{i*}) + \sum_{i=1}^{N} \lambda_{\text{score}} \mathcal{L}_{score}(s_i) + \lambda_{\text{reg}} e^{-\gamma t} \mathcal{L}_{mask}(m_i)$$

where $\lambda_{\text{score}} = 0.05$, $\lambda_{\text{reg}} = 0.1$ weigh the losses, $\gamma = 0.2$ is decay rate, $t$ is current epoch, and $N$ is the number of prediction branches.

## 4 DATASET

Unlike other dense prediction tasks, scaling SOD datasets poses unique challenges that cannot be addressed by simply leveraging existing collections such as LAION (Schuhmann et al., 2022). SOD requires samples with distinct foreground objects, and annotation demands significant expertise and attention to detail, particularly for high-resolution images with precise boundary requirements. These constraints make traditional manual dataset curation both impractical and cost-inefficient. Our goal is to generate large-scale synthetic data that accurately reflects real-world distributions.

### 4.1 MULTI-MODAL DATASET DIFFUSION

Large-scale diffusion transformers, such as FLUX (Labs, 2023), with 12B parameters encode rich semantic and spatial representations during the generation process. Rather than ignoring these latent representations and relying on teacher models that predict masks directly from generated images, we extend the diffusion model to output masks by combining multiple complementary modalities. We extract latent feature maps that encode spatial layout understanding, concept-attention maps

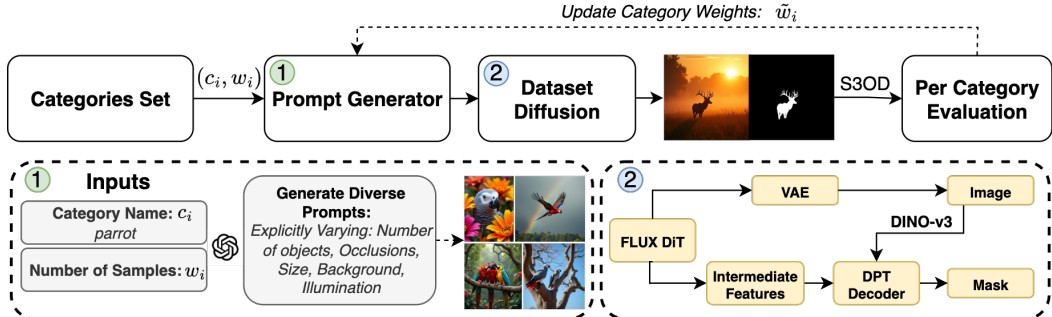

Figure 3: **Iterative Generation Pipeline.** The LLM (Achiam et al., 2023) generates a library of diverse prompts for a large set of object categories. These prompts guide a diffusion model to generate synthetic images with corresponding masks. The resulting dataset is used to train the SOD model, which is evaluated category-wise. Performance feedback from a trained SOD model dynamically adjusts the category weights $\tilde{w}_i$, prioritizing challenging cases in subsequent iterations.

that provide interpretable semantic localization, and DINO-v3 features from the decoded images that capture fine-grained visual semantics. This multi-modal supervision mitigates data scarcity and ensures alignment between generated images and their corresponding masks.

**DiT Feature Maps.** FLUX DiT employs a hybrid architecture with 19 dual-stream transformer blocks (processing text and image tokens separately) and 38 single-stream blocks (operating on concatenated sequences). We extract feature maps from four single-stream transformer blocks at layers $\{4, 16, 27, 36\}$, encoding multi-scale spatial representations across generation stages. Each block outputs features $\mathbb{R}^{B \times (L_T + L_I) \times 3072}$ where $L_T = 512$. We extract only image tokens, $\mathbb{R}^{B \times L_I \times 3072}$, and project them to 768 dimensions via learned projections. These features encode the model's internal spatial understanding used during generation.

**Concept Attention Maps.** Common dataset generation methods (Wu et al., 2023a) extract mean attention maps across all text tokens, producing semantically ambiguous supervision. Instead, we use a static set of concepts to obtain interpretable, consistent maps. Following the concept-attention framework (Helbling et al., 2025), for each generated image, we compute attention maps between image patches and static concept tokens. For concept token $c$ and image patch $x$, we compute:

$$A_{concept}(x, y) = \text{softmax}(o_x \cdot o_c^T)$$

where $o_x$ and $o_c$ are attention output vectors from the multi-modal transformer layers. For each sample, we extract two concept attention maps using the primary object category (e.g., "dog") and the "background" token, yielding interpretable maps $\{A_{object}, A_{background}\}$ that consistently encode object locations and background regions.

**DINO-v3 Visual Features.** We extract semantic visual features from generated images using DINO-v3 (ViT-L), providing rich object-level representations that capture fine-grained visual semantics through self-supervised learning trained on large-scale real-world data.

The three modalities are fused through a dedicated module that projects each to a common 256-dimensional space via separate convolutional branches with batch normalization. FLUX features and concept maps are upsampled to match DINO-v3 resolution using bilinear interpolation. The projected features are concatenated channel-wise and processed through a two-stage convolutional network ($3 \times 3$ followed by $1 \times 1$ convolution), with the result residually combined with the original DINO-v3 features to produce unified multi-modal representations. We feed this combined representation into the DPT decoder, supervising it with DIS-5K, HR-SOD, UHRSOD, and DUTS datasets, ensuring the model learn how to decode multiple sources into a fine-grained segmentation mask.

## 4.2 ITERATIVE DATA SYNTHESIS

To incorporate a feedback mechanism into the data generation, we introduce an iterative process that adjusts generation parameters based on the downstream model's performance for subsequent rounds. After training the model on synthetic data $\mathcal{D}^{(r)}$, we evaluate its performance on a held-out test set for each category $c_i$. For each image $I_j$, we compute a score $\kappa(I_j)$, which is the average IoU score across

Table 1: **SOD Datasets Statistics:** S3OD dataset is orders of magnitudes larger than existing datasets and contains a wide variety of scenes and objects.

| Metric | DUTS | ECSSD | HKU-IS | DUT-OMRON | UHRSD | HRSOD | DIS-5K | S3OD (ours) |
|---|---|---|---|---|---|---|---|---|
| # of Images | 15,570 | 1,000 | 4,447 | 5,168 | 5,920 | 2,010 | 5,000 | 139,981 |
| # of Unique Objects | 1152 | 310 | 551 | 749 | 948 | 381 | 758 | 1676 |

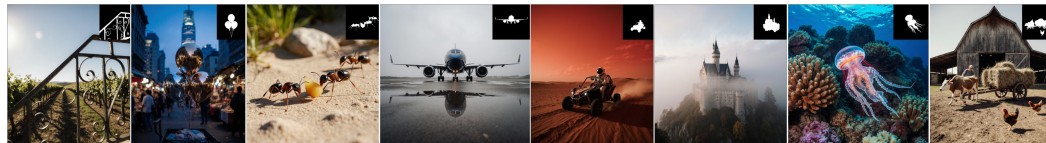

Figure 4: **S3OD Dataset:** The dataset consists of diverse object categories and complex scenes that closely reflect real-world environments, featuring various lighting conditions, spatial compositions, and object interactions. All samples are generated with multi-modal dataset diffusion.

various image transformations (flipping, etc.). $\kappa(I_j)$ is high if the prediction is consistent across augmentations. We then compute a mean category score $\bar{\kappa}_i$ by averaging these scores across all images in category $c_i$. The category weights $w_i^{(r+1)}$ for the next iteration are updated proportionally to the inverse of these scores, ensuring categories with lower performance receive more samples in subsequent generations. Specifically, we map the category scores through a non-linear scaling function: $w_i^{(r+1)} = w_{\min} + w_{\text{new}} e^{-\alpha(\bar{\kappa}_i - \beta)}$, where $\alpha = 8$ and $\beta = 0.5$ control the strength of the performance-based skew, $w_{\min} = \frac{1}{|\mathcal{C}|}$ is a minimum weight per class, and $w_{\text{new}} = \frac{4}{|\mathcal{C}|}$ is the maximal possible over-weighting. This scales up the weights of categories with scores below a given threshold while maintaining a minimum weight for well-performing categories. This adaptive sampling strategy ensures that the synthetic data generation process continuously evolves, producing examples that maximize model improvement. The pipeline is visualized in Figure 3.

### 4.3 Multi-Stage Quality Filtering

While synthetic data generation offers scalability, it inevitably produces imperfect samples that can degrade training quality. To ensure high-quality datasets, we implement a comprehensive multi-stage filtering pipeline that addresses standard failure modes in synthetic data generation.

**Consistency Filtering.** We evaluate prediction consistency using a separate large model trained without FLUX features. For each sample, we compute the IoU between the original prediction and the horizontally flipped prediction, filtering samples with an IoU below the $\tau = 0.8$ consistency threshold. Low consistency scores often indicate overly ambiguous samples where even robust models struggle to maintain coherent predictions, suggesting fundamental issues with the generated image-mask pairs.

**Mask Quality Assessment.** We employ a Gemma-3 VLM (Team et al., 2025) to evaluate mask quality, identifying severe artifacts such as fragmentation, noise, or artifacts that commonly occur in image segmentation. Only masks with cohesive white regions ($\leq 5$ main components) pass this stage, ensuring clean supervision signals for model training.

**Semantic Validation.** In a second pass, the Gemma VLM evaluates semantic correctness by analyzing the original image and the mask overlay. This stage ensures both the presence of clear, salient objects and adequate mask coverage ($> 70\%$ of the main object), thereby filtering out samples in which the multi-modal supervision fails to capture the intended semantic content.

This multi-stage approach removes 6.8% of the generated samples, significantly improving dataset quality while preserving the scale advantages of manual annotation.

### 4.4 Implementation Details

We generate 139,981 high-resolution data samples Figure 4 in three rounds ($R = 3$) which is **131%** more than 11 most common academic benchmarks combined. We sample category names from the ImageNet taxonomy, which covers a wide range of objects and activities Table 1. The first round generates 100 images per category, followed by a second and third rounds with an additional 25,000 images in each, prioritizing challenging categories. During processing, 6.8% of the samples are

filtered out. The images are generated with a FLUX model with 25 inference steps. For each image, we randomly sample an aspect ratio from a fixed set of common image resolutions, further expanding the dataset variety. All student models are trained using the ViT-B backbone (Dosovitskiy et al., 2020). Model training on S3OD dataset takes 2 days on 8 H200 GPUs.

## 5 EXPERIMENTAL EVALUATION

We evaluate dataset and model generalization, as well as performance on various benchmarks.

### 5.1 EVALUATION PROTOCOL

The performance of the salient object detection models is evaluated on six datasets of two domains. For dichotomous image segmentation (DIS), we use DIS-5K (Qin et al., 2022), which contains 5,470 high-resolution images with extremely fine-grained labels for camouflaged, salient, and meticulous objects across varied backgrounds. For Salient Object Detection (HR-SOD), we evaluate on three high-resolution benchmarks: UHRSD (Xie et al., 2022b) (5,920 images at 4K-8K resolution), HRSOD-TE (Zeng et al., 2019) (400 test images with shortest edge >1200 pixels), and DAVIS-S (92 images from DAVIS (Pont-Tuset et al., 2017) video segmentation dataset). We also include two low-resolution benchmarks: DUT-OMRON (Yang et al., 2013) (5,168 images with complex backgrounds) and DUTS-TE (Wang et al., 2017) (5,019 test images from the largest available SOD dataset). All datasets include pixel-wise ground-truth annotations for quantitative evaluation.

**Metrics.** We evaluate each model with commonly used metrics: maximum F-measure ($F_{1max}$) (Achanta et al., 2009), Mean Average Error (MAE) (Perazzi et al., 2012), structure measure ($S_\alpha$) (Fan et al., 2017) and enhanced alignment measure ($E_M^\Phi$) (Fan et al., 2018). The F-measure ($F_\beta$) balances precision and recall, computed with $\beta^2 = 0.3$ to emphasize precision. MAE computes the average absolute difference between the predicted and ground-truth masks. The structure measure $S_\alpha$ evaluates preservation of object-aware ($S_o$) and region-aware ($S_r$) structural similarities, computed as $S_m = \alpha * S_o + (1 - \alpha) * S_r$ with $\alpha = 0.5$. The enhanced alignment measure $E_M^\Phi$ combines local and global similarity information, jointly capturing image-level statistics and pixel-level matching.

Table 2: **Cross-Dataset Generalization:** S3ODNet trained on synthetic data only demonstrates superior generalization across all datasets comparing to other methods trained on subtasks datasets. SOD datasets stand for (HRSOD-TR (Zeng et al., 2019), UHRSD-TR (Xie et al., 2022b) and DUTS-TR (Wang et al., 2017)). **Best** and second best results highlighted.

| Method | Data | DIS-1 $F_m \uparrow$ | $S_\alpha \uparrow$ | $E_M^\Phi \uparrow$ | MAE$\downarrow$ | DIS-2 $F_m \uparrow$ | $S_\alpha \uparrow$ | $E_M^\Phi \uparrow$ | MAE$\downarrow$ | DIS-3 $F_m \uparrow$ | $S_\alpha \uparrow$ | $E_M^\Phi \uparrow$ | MAE$\downarrow$ | DIS-4 $F_m \uparrow$ | $S_\alpha \uparrow$ | $E_M^\Phi \uparrow$ | MAE$\downarrow$ | Overall $F_m \uparrow$ | $S_\alpha \uparrow$ | $E_M^\Phi \uparrow$ | MAE$\downarrow$ |
|---|---|---|---|---|---|---|---|---|---|---|---|---|---|---|---|---|---|---|---|---|---|
| InSpyreNet | DUTS | .786 | .822 | .857 | .064 | .828 | .845 | .877 | .057 | .839 | .848 | .883 | .059 | .789 | .806 | .838 | .082 | .811 | .830 | .864 | .065 |
| BiRefNet | SOD | .812 | .841 | .863 | .049 | .844 | .855 | .877 | .050 | .855 | .856 | .881 | .053 | .790 | .803 | .824 | .081 | .825 | .839 | .861 | .058 |
| S3ODNet | SOD | .850 | .885 | .902 | .046 | .880 | .870 | .914 | .043 | .888 | .875 | .928 | .040 | .833 | .823 | .881 | .069 | .863 | .856 | .906 | .049 |
| S3ODNet | S3OD | **.865** | **.884** | **.917** | **.034** | **.896** | **.898** | **.933** | **.032** | **.901** | **.895** | **.938** | **.033** | **.861** | **.857** | **.913** | **.054** | **.881** | **.884** | **.925** | **.039** |

| Method | Data | DAVIS-S $F_m \uparrow$ | $S_\alpha \uparrow$ | $E_M^\Phi \uparrow$ | MAE$\downarrow$ | HRSOD-TE $F_m \uparrow$ | $S_\alpha \uparrow$ | $E_M^\Phi \uparrow$ | MAE$\downarrow$ | UHRSOD-TE $F_m \uparrow$ | $S_\alpha \uparrow$ | $E_M^\Phi \uparrow$ | MAE$\downarrow$ | DUTS-TE $F_m \uparrow$ | $S_\alpha \uparrow$ | $E_M^\Phi \uparrow$ | MAE$\downarrow$ | DUT-OMRON $F_m \uparrow$ | $S_\alpha \uparrow$ | $E_M^\Phi \uparrow$ | MAE$\downarrow$ |
|---|---|---|---|---|---|---|---|---|---|---|---|---|---|---|---|---|---|---|---|---|---|
| InSpyreNet | DIS | .921 | .937 | .966 | .015 | .891 | .912 | .923 | .038 | .914 | .922 | .932 | .033 | .845 | .880 | .895 | .046 | .713 | .801 | .812 | .071 |
| BiRefNet | DIS | .919 | .936 | .961 | .014 | .887 | .915 | .926 | .031 | .922 | .924 | .937 | .032 | .860 | .886 | .910 | .036 | .744 | .819 | .835 | .054 |
| MVANet | DIS | .907 | .929 | .959 | .016 | .902 | .919 | .930 | .033 | .922 | .926 | .941 | .032 | .852 | .877 | .893 | .042 | .711 | .792 | .838 | .072 |
| S3ODNet | DIS | .951 | .950 | .973 | .010 | .923 | .913 | .932 | .030 | .946 | .927 | .947 | .029 | .902 | .901 | .926 | .035 | .808 | .830 | .858 | .061 |
| S3ODNet | S3OD | **.970** | **.967** | **.988** | **.005** | **.954** | **.955** | **.972** | **.016** | **.954** | **.944** | **.961** | **.023** | **.937** | **.938** | **.962** | **.020** | **.860** | **.887** | **.911** | **.040** |

### 5.2 CROSS-DATASET GENERALIZATION

We argue that the most important aspect of modern salient object segmentation models should be generalizing to new image distributions. We evaluate cross-task generalization by training the model on the DIS-5K (Qin et al., 2022) dataset and evaluating on SOD benchmarks, and vice versa. A robust, generalizable method is expected to perform well across all benchmark datasets, since they all focus on the same high-level problem. The results are presented in Table 2. S3OD trained on a combination of SOD datasets achieves superior generalization compared to BiRefNet (Zheng et al., 2024), or InSpyreNet (Kim et al., 2022).

Remarkably, even training solely on synthetic data enables the method to achieve state-of-the-art generalization, reducing the MAE compared to the model trained on DIS-5K by **50.0%**, **46.7%**, **20.7%**, **42.9%**, and **34.4%**. The models trained on DIS-5K only (3000 images) and evaluated on SOD benchmarks all achieve comparable results, demonstrating the importance of data scale and

| Method | Data | DAVIS-S | | | | HRSOD-TE | | | | UHRSOD-TE | | | | DUTS-TE | | | | DUT-OMRON | | | |
|---|---|---|---|---|---|---|---|---|---|---|---|---|---|---|---|---|---|---|---|---|---|
| | | $F_m$ ↑ | $S_\alpha$ ↑ | $E_M^\Phi$ ↑ | MAE↓ | $F_m$ ↑ | $S_\alpha$ ↑ | $E_M^\Phi$ ↑ | MAE↓ | $F_m$ ↑ | $S_\alpha$ ↑ | $E_M^\Phi$ ↑ | MAE↓ | $F_m$ ↑ | $S_\alpha$ ↑ | $E_M^\Phi$ ↑ | MAE↓ | $F_m$ ↑ | $S_\alpha$ ↑ | $E_M^\Phi$ ↑ | MAE↓ |
| MVANet | DIS | .907 | .929 | .959 | .016 | .902 | .919 | .930 | .033 | .922 | .926 | .941 | .032 | .852 | .877 | .893 | .042 | .711 | **.792** | .838 | .072 |
| MVANet | S3OD | .951 | .958 | .975 | .008 | .950 | .948 | .954 | .019 | .951 | .943 | .942 | .024 | .875 | .893 | .901 | .039 | .776 | .791 | .873 | .064 |
| BiRefNet | DIS | .919 | .936 | .961 | .014 | .887 | .915 | .926 | .031 | .922 | .924 | .937 | .032 | .860 | .886 | .910 | .036 | .744 | .819 | .835 | .054 |
| BiRefNet | S3OD | .963 | .958 | .978 | .009 | .956 | .951 | .965 | .019 | .955 | .949 | .962 | .022 | .928 | .931 | .951 | .024 | .845 | .882 | .899 | .045 |
| S3ODNet | DIS | .951 | .950 | .973 | .010 | .923 | .913 | .932 | .030 | .946 | .927 | .947 | .029 | .902 | .901 | .926 | .035 | .808 | .830 | .858 | .061 |
| S3ODNet | S3OD | **.970** | **.967** | **.988** | **.005** | **.954** | **.955** | **.972** | **.016** | **.954** | **.944** | **.961** | **.023** | **.937** | **.938** | **.962** | **.020** | **.860** | **.887** | **.911** | **.040** |

Table 3: Impact of S3OD dataset on salient object detection performance across different methods. Training on S3OD improves generalization across all methods.

the impact of overfitting on subtask-specific performance. Still, S3OD trained on synthetic data demonstrates strong generalization across all benchmarks.

To further validate the impact of S3OD dataset, we retrained BiReftNet (Zheng et al., 2024) and MVANet (Yu et al., 2024) on our synthetic data. Results are reported in Table 3 and are consistent with other evaluations. Training on S3OD improves the generalization of all models, and S3ODNet still outperforms other methods trained in the same setup.

## 5.3 STATE-OF-THE-ART COMPARISON

Table 4: Quantitative comparison on DIS5K and SOD benchmarks. Best results highlighted in **bold**. The S3ODNet $^*$ are the metrics computed with the best match over three predicted masks.

| Method | DIS-1 | | | | DIS-2 | | | | DIS-3 | | | | DIS-4 | | | | Overall | | | |
|---|---|---|---|---|---|---|---|---|---|---|---|---|---|---|---|---|---|---|---|---|
| | $F_m$ ↑ | $S_\alpha$ ↑ | $E_M^\Phi$ ↑ | MAE↓ | $F_m$ ↑ | $S_\alpha$ ↑ | $E_M^\Phi$ ↑ | MAE↓ | $F_m$ ↑ | $S_\alpha$ ↑ | $E_M^\Phi$ ↑ | MAE↓ | $F_m$ ↑ | $S_\alpha$ ↑ | $E_M^\Phi$ ↑ | MAE↓ | $F_m$ ↑ | $S_\alpha$ ↑ | $E_M^\Phi$ ↑ | MAE↓ |
| SAM-HQ | **.897** | **.907** | **.943** | **.019** | .889 | .883 | .928 | .029 | .851 | .851 | .897 | .045 | .763 | .799 | .843 | .088 | .850 | .860 | .903 | .045 |
| InSpyreNet | .845 | .873 | .874 | .043 | .894 | .905 | .916 | .036 | .919 | .918 | .940 | .034 | .905 | **.905** | .936 | .042 | .891 | .900 | .917 | .039 |
| BiRefNet | .860 | .885 | .911 | .037 | .894 | .900 | .930 | .036 | .925 | .919 | .955 | .028 | .904 | .900 | .939 | .039 | .896 | .901 | .934 | .035 |
| MVANet | .862 | .880 | .906 | .039 | .909 | .912 | .942 | .032 | .924 | .918 | .954 | .030 | .907 | **.905** | .946 | .039 | .900 | .904 | .937 | .035 |
| S3ODNet | .892 | .902 | .932 | .031 | **.923** | **.921** | **.953** | **.026** | **.930** | **.920** | **.960** | **.025** | **.909** | .902 | **.954** | **.034** | **.914** | **.911** | **.950** | **.029** |
| S3ODNet $^*$ | .916 | .924 | .960 | .018 | .941 | .936 | .973 | .016 | .941 | .931 | .975 | .018 | .914 | .907 | .967 | .027 | .928 | .924 | .969 | .020 |

| Method | DAVIS-S | | | | HRSOD-TE | | | | UHRSD-TE | | | | DUTS-TE | | | | DUT-OMRON | | | |
|---|---|---|---|---|---|---|---|---|---|---|---|---|---|---|---|---|---|---|---|---|
| | $F_m$ ↑ | $S_\alpha$ ↑ | $E_M^\Phi$ ↑ | MAE↓ | $F_m$ ↑ | $S_\alpha$ ↑ | $E_M^\Phi$ ↑ | MAE↓ | $F_m$ ↑ | $S_\alpha$ ↑ | $E_M^\Phi$ ↑ | MAE↓ | $F_m$ ↑ | $S_\alpha$ ↑ | $E_M^\Phi$ ↑ | MAE↓ | $F_m$ ↑ | $S_\alpha$ ↑ | $E_M^\Phi$ ↑ | MAE↓ |
| InSpyreNet | .977 | .973 | .987 | .007 | .956 | .956 | .962 | .018 | .957 | .953 | .965 | .020 | .932 | .936 | .956 | .024 | .823 | .872 | .906 | .046 |
| BiRefNet | **.979** | **.975** | .989 | .006 | **.963** | .957 | .973 | .016 | .963 | **.957** | .969 | **.016** | .943 | .944 | .962 | .018 | .839 | .882 | .896 | .038 |
| S3ODNet | **.979** | .974 | **.993** | **.004** | **.963** | **.961** | **.978** | **.013** | **964** | .952 | **.969** | .018 | **.954** | **.949** | **.972** | **.015** | **.879** | **.898** | **.924** | **.032** |
| S3ODNet $^*$ | .982 | .977 | .993 | .004 | .979 | .973 | .991 | .005 | .977 | .966 | .985 | .008 | .963 | .959 | .987 | .008 | .907 | .919 | .953 | .023 |

Prior work does not evaluate cross-task generalization and trains task/benchmark-specific models. While we argue that the evaluation above is the way forward for salient object segmentation, we also evaluate in the historically used setting. We finetune the model trained on our S3OD dataset on both the DIS-5K (Qin et al., 2022) and a combination of SOD datasets (HR-SOD (Zeng et al., 2019), UHRSOD (Xie et al., 2022b), DUTS-TR (Wang et al., 2017)). We report the results in Table 4. S3OD significantly outperforms all the other methods on DIS-5K benchmarks achieving a new state-of-the-art and reducing the error rate by **14.0%**, **7.3%**, **20.6%** and **17.1%**.

However, salient object detection benchmarks have become highly saturated. S3OD achieves superior results on HRSOD-TE (Zeng et al., 2019), DUTS-TE (Wang et al., 2017), and DUT-OMRON (Yang et al., 2013), even though all models are trained on the first two datasets. The evaluation on the DUT-OMRON benchmark serves as the strongest generalization test as none of the models were trained or fine-tuned on it, and the benchmark consists of 5,168 samples. S3OD achieves **24.8%**, **13.6%**, **26.9%** and **15.8%** reduction in error rate compared to BiRefNet. Notably, on UHRSD (Xie et al., 2022b), the largest HR-SOD training dataset, and DAVIS-S, which contains only 92 images, all large models with transformer backbones achieve comparable results. This is another indicator of benchmark saturation and supports our choice of cross-task generalization evaluation. The variant S3OD $^*$ computes the metrics using the best match among the three masks and the ground-truth mask. This oracle evaluation uses ground-truth information and cannot be compared with other methods. However, it demonstrates the inherent ambiguity in the data annotations and/or the task. This confirms that ambiguity-aware modelling will be highly useful in practical applications.

## 5.4 SYNTHETIC DATA EVALUATION

We also evaluate our data generation mechanism compared to other data synthesis methods. We measure the impact of synthetic data on performance and generalization, evaluating S3OD and other

Table 5: **Synthetic Data Generation Evaluation:** S3ODNet model is trained on a combination of DIS-5K and 3 synthetic datasets. Training with S3OD dataset significantly improves generalization.

| Training Data | DIS (1-4) | | | | HRSOD-TE | | | | UHRSD-TE | | | | DUTS-TE | | | | DUT-OMRON | | | |
|---|---|---|---|---|---|---|---|---|---|---|---|---|---|---|---|---|---|---|---|---|
| | $F_m \uparrow$ | $S_\alpha \uparrow$ | $E_M^\Phi \uparrow$ | MAE↓ | $F_m \uparrow$ | $S_\alpha \uparrow$ | $E_M^\Phi \uparrow$ | MAE↓ | $F_m \uparrow$ | $S_\alpha \uparrow$ | $E_M^\Phi \uparrow$ | MAE↓ | $F_m \uparrow$ | $S_\alpha \uparrow$ | $E_M^\Phi \uparrow$ | MAE↓ | $F_m \uparrow$ | $S_\alpha \uparrow$ | $E_M^\Phi \uparrow$ | MAE↓ |
| DIS | .910 | .897 | .943 | .032 | .923 | .913 | .932 | .032 | .946 | .927 | .947 | .030 | .902 | .901 | .925 | .036 | .808 | .830 | .858 | .061 |
| DIS + MaskFactory | .912 | .904 | **.950** | **.030** | .910 | .916 | .936 | .031 | .937 | .926 | .947 | .030 | .886 | .898 | .924 | .038 | .774 | .812 | .842 | .071 |
| DIS + DatasetDM | .898 | .889 | .939 | .036 | .899 | .896 | .911 | .041 | .932 | .914 | .934 | .037 | .872 | .877 | .900 | .048 | .770 | .795 | .818 | .080 |
| DIS + S3OD | .908 | **.905** | .945 | **.030** | **.944** | **.946** | **.963** | **.020** | **.950** | **.940** | **.958** | **.024** | **.924** | **.928** | **.951** | **.025** | **.842** | **.871** | **.899** | **.048** |

synthetic data generation methods (Wu et al., 2023a; Qian et al., 2024). MaskFactory (Qian et al., 2024) augments the DIS-5K (Qin et al., 2022) dataset with both rigid and non-rigid transforms and generates a new set of images conditioned on augmented masks. To ensure fair comparison, we train our model on DIS-5K and a mix of DIS-5K and three synthetic datasets. Since the other two synthetic datasets contain only 10,000 train images, we also subsample a subset from S3OD of the same size from the 2nd iteration of data generation. We evaluate the model both on DIS and SOD benchmarks. The results are presented in Table 5.

**Results.** Interestingly, S3OD achieves comparable performance to MaskFactory (Qian et al., 2024) on the DIS-5K test set, even though it was not fine-tuned for categories and types of object in this benchmark, despite MaskFactory utilising the DIS-5K train set to generate augmented masks. On the other four SOD benchmarks, S3OD demonstrates significantly stronger generalization and performance compared to both the original training dataset and other synthetic data generation methods, thereby demonstrating the diversity and versatility of our data generation method.

## 5.5 GENERALIZATION TO CAMOUFLAGED OBJECT DETECTION

| Method | Data | COD10K | | | | CAMO | | | | NC4K | | | |
|---|---|---|---|---|---|---|---|---|---|---|---|---|---|
| | | $F_m \uparrow$ | $S_\alpha \uparrow$ | $E_M^\Phi \uparrow$ | MAE↓ | $F_m \uparrow$ | $S_\alpha \uparrow$ | $E_M^\Phi \uparrow$ | MAE↓ | $F_m \uparrow$ | $S_\alpha \uparrow$ | $E_M^\Phi \uparrow$ | MAE↓ |
| S3ODNet | SOD | .850 | .862 | .911 | .034 | .858 | .848 | .893 | .061 | .896 | .889 | .929 | .034 |
| S3ODNet | DIS | .832 | .853 | .896 | .035 | .845 | .846 | .892 | .058 | .885 | .882 | .922 | .035 |
| S3ODNet | MaskFactory | .809 | .828 | .884 | .035 | .849 | .838 | .889 | .060 | .872 | .864 | .909 | .038 |
| S3ODNet | S3OD | **.854** | **.880** | **.920** | **.033** | **.859** | **.864** | **.906** | **.056** | **.897** | **.901** | **.936** | **.032** |
| FSPNet | COD | .769 | .851 | .895 | .026 | .830 | .856 | .899 | .050 | .843 | .879 | .915 | .035 |
| BiRefNet | COD | .888 | .913 | .960 | .014 | .904 | **.904** | **.954** | **.030** | .909 | .914 | .953 | .023 |
| S3ODNet | S3OD +COD | **.911** | **.923** | **.970** | **.012** | **.908** | .903 | .949 | .031 | **.923** | **.920** | **.961** | **.020** |

Table 6: **Evaluation on COD benchmarks.** We evaluate generalization to Camouflaged Object Detection. When trained on S3OD dataset S3ODNet reach the strongest generalization to the new task in zero-shot transfer setting, comparing to other real and synthetic dataset. Fine-tuned on COD data S3ODNet achieves state-of-the-art results on COD-10K and NC4K benchmarks.

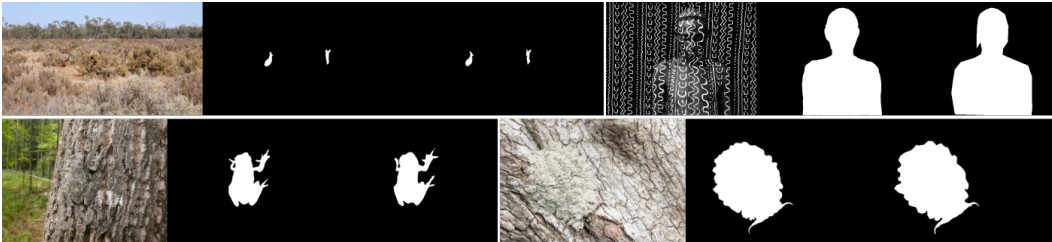

Figure 5: **Zero-shot Evaluation on Camouflaged Object Detection.** Left to Right: Image, Predicted Mask, Ground Truth. Our model trained on S3OD generalizes to detecting camouflaged objects despite being trained exclusively on synthetic SOD data.

To evaluate the generalization of our dataset and model beyond salient object detection, we assess transfer to Camouflaged Object Detection (Fan et al., 2020): a challenging task in which objects are specifically designed to blend with their backgrounds. We evaluate on three COD benchmarks: COD10K (Fan et al., 2020), CAMO (Le et al., 2019), and NC4K (Lv et al., 2021). Table 6 shows that S3ODNet trained solely on S3OD (without any real data) achieves strong zero-shot performance, outperforming models trained on SOD, DIS, or MaskFactory datasets across all metrics. Next, following the BiRefNet setup, and finetune S3OD on CAMO and COD-10K train sets. The finetuned model reaches state-of-the-art results on COD10K ($F_m = 0.911$ vs BiRefNet's 0.888) and NC4K

($F_m = 0.923$ vs $0.909$). Similar to SOD evaluation, we observe that the smallest benchmark the other models are also trained on (CAMO with only 250 test images), shows saturation due to overfitting. This validates that our synthetic data teaches generalizable segmentation principles beyond salient object detection. Interestingly, S3ODNet, trained only on our synthetic data, outperforms some methods trained on COD datasets (Huang et al., 2023).

Figure 5 visualizes predictions on challenging camouflaged scenes, showing that models trained only on S3OD successfully detect occluded complex objects with ambiguous boundaries, confirming that even such challenging scenarios are represented in S3OD synthetic dataset.

## 5.6 ABLATION STUDY

Table 7: **Iterative Data Generation Ablation:** Progressively generating hard samples improves model performance and generalization across all datasets.

| Training Data | DIS (1-4) | | | | HRSOD-TE | | | | DUTS-TE | | | | DUT-OMRON | | | |
|---|---|---|---|---|---|---|---|---|---|---|---|---|---|---|---|---|
| | $F_m \uparrow$ | $S_\alpha \uparrow$ | $E_M^\Phi \uparrow$ | MAE$\downarrow$ | $F_m \uparrow$ | $S_\alpha \uparrow$ | $E_M^\Phi \uparrow$ | MAE$\downarrow$ | $F_m \uparrow$ | $S_\alpha \uparrow$ | $E_M^\Phi \uparrow$ | MAE$\downarrow$ | $F_m \uparrow$ | $S_\alpha \uparrow$ | $E_M^\Phi \uparrow$ | MAE$\downarrow$ |
| S3OD Single Round | .879 | .883 | .916 | .041 | .951 | .953 | .969 | .018 | .933 | .935 | .959 | **.020** | .855 | .881 | .907 | .042 |
| S3OD (2 rounds) | .880 | **.884** | .918 | .040 | .953 | .954 | .971 | .017 | .935 | **.939** | .961 | **.020** | .859 | .885 | .908 | **.040** |
| S3OD (3 rounds) | **.881** | **.884** | **.925** | **.039** | **.954** | **.955** | **.972** | **.016** | **.937** | .938 | **.962** | **.020** | **.860** | **.887** | **.911** | **.040** |

Table 8: **Data Diffusion Model Ablation:** Combining all three modalities achieves optimal performance across benchmarks.

| DINO-v3 | DiT Maps | Concept Maps | DIS (1-4) | | | | HRSOD-TE | | | | UHRSD-TE | | | | DUTS-TE | | | | DUT-OMRON | | | |
|---|---|---|---|---|---|---|---|---|---|---|---|---|---|---|---|---|---|---|---|---|---|---|---|
| | | | $F_m \uparrow$ | $S_\alpha \uparrow$ | $E_M^\Phi \uparrow$ | MAE$\downarrow$ | $F_m \uparrow$ | $S_\alpha \uparrow$ | $E_M^\Phi \uparrow$ | MAE$\downarrow$ | $F_m \uparrow$ | $S_\alpha \uparrow$ | $E_M^\Phi \uparrow$ | MAE$\downarrow$ | $F_m \uparrow$ | $S_\alpha \uparrow$ | $E_M^\Phi \uparrow$ | MAE$\downarrow$ | $F_m \uparrow$ | $S_\alpha \uparrow$ | $E_M^\Phi \uparrow$ | MAE$\downarrow$ |
| ✗ | ✓ | ✓ | .710 | .743 | .783 | .091 | .733 | .784 | .789 | .097 | .865 | .868 | .890 | .054 | .773 | .805 | .840 | .070 | .681 | .733 | .772 | .095 |
| ✓ | ✗ | ✓ | .913 | .909 | .949 | .029 | .959 | .958 | .974 | .014 | .965 | .953 | **.971** | .017 | **.950** | .945 | .968 | .017 | .870 | .887 | .911 | .036 |
| ✓ | ✓ | ✗ | .914 | .906 | .944 | .030 | .961 | .957 | .972 | .014 | .965 | .952 | **.971** | .016 | .949 | .943 | .966 | .017 | .871 | .889 | .915 | .036 |
| ✓ | ✓ | ✓ | **.917** | **.913** | **.951** | **.028** | **.962** | **.961** | **.976** | **.012** | **.966** | **.953** | **.971** | **.016** | .948 | **.944** | **.969** | **.016** | **.873** | **.891** | **.918** | **.034** |

Table 9: **Architecture Ablation:** Multi-mask decoder improves performance on DIS-5K.

| Backbone | $N_M$ | $F_m \uparrow$ | $S_\alpha \uparrow$ | MAE$\downarrow$ |
|---|---|---|---|---|
| Swin-B | 1 | .884 | .883 | .044 |
| DINO-v3 | 1 | .909 | .911 | .033 |
| DINO-v3 | 2 | .892 | .896 | .034 |
| DINO-v3 | 3 | **.914** | **.913** | **.031** |

Table 10: **Prompt Generator:** LLM prompts improve diversity and quality.

| Prompt | CLIP$\uparrow$ | IS$\uparrow$ |
|---|---|---|
| Class Name | .399 | 67.8 |
| GPT | **.434** | **.98.1** |

We evaluate our multi-modal data diffusion approach and architectural components. Table 8 shows individual feature types are insufficient: diffusion features alone cannot decode high-resolution masks, while DINO-v3, despite strong performance, can suffer from train-test distribution gaps when applied to generated images. The combination of all three modalities achieves optimal performance across benchmarks, with diffusion features providing crucial complementary information for challenging, ambiguous cases. The DINO-v3 backbone significantly outperforms Swin-B (Table 9), demonstrating the value of foundation models. Three mask predictions also yield the best performance, proving multi-mask effectiveness. Iterative generation Table 7 consistently improves performance with 3.6% F-measure gain on DIS datasets and 5.3% on DUT-OMRON, confirming the effectiveness of prioritizing challenging categories. LLM-generated prompts improve synthetic image quality with 44.7% Inception Score Table 10 increase over simple class names.

## 6 CONCLUSION

We demonstrate that combining features from generative and discriminative models: diffusion transformer feature maps, concept attention maps, and DINO-v3 features enables effective synthetic data generation for salient object detection. Our iterative generation framework dynamically prioritizes challenging categories, while the ambiguity-aware architecture naturally handles multiple valid interpretations. This pipeline significantly improves cross-dataset generalization and provides a scalable framework for addressing data scarcity in dense prediction tasks, suggesting that synthetic datasets can complement manual annotations in computer vision applications.

ACKNOWLEDGMENTS

This research was supported by ERC StG 101222037-Volute, AIST policy-based budget project "R&D on Generative AI Foundation Models for the Physical Domain", and Japan Science and Technology Agency (JST) as part of Adopting Sustainable Partnerships for Innovative Research Ecosystem (ASPIRE), Grant Number JPMJAP2518. Computational resources were provided by ABCI 3.0 from AIST and AIST Solutions. We are grateful to Andrew Zisserman for valuable discussions and feedback that helped improve this work.

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

## A   FEATURE COMPLEMENTARITY

Figure 6 visualizes the three feature sources on dataset samples. Concept attention maps provide strong but coarse foreground-background separation through explicit semantic grounding. DINO-v3 features capture fine-grained visual semantics where similar regions exhibit similar embeddings, enabling strong object-level understanding. FLUX DiT features encode spatial scene parsing information from the generative process, including boundary localization and structural composition. The visualization also demonstrates the limitations of individual feature sources. Concept maps provide strong foreground cues in a simple scene but fail to precisely localize foreground objects in more complex scenarios (rows 3 and 4) – demonstrating the limitation of unsupervised segmentation methods that rely only on attention maps (Helbling et al., 2025). Rows 3 and 5 also show DiT feature maps capabilities: in contrast to DINO features, the objects in reflection or snow pile patches have higher similarity as the diffusion model efficiently reuses the information during generation. This precisely demonstrates the importance of combining multiple feature sources: in highly complex, ambiguous scenes, generative and discriminative features complement each other, allowing for the decoding of a high-quality mask. Note that FLUX and DINO-v3 features are high-dimensional and visualized via PCA for interpretability.

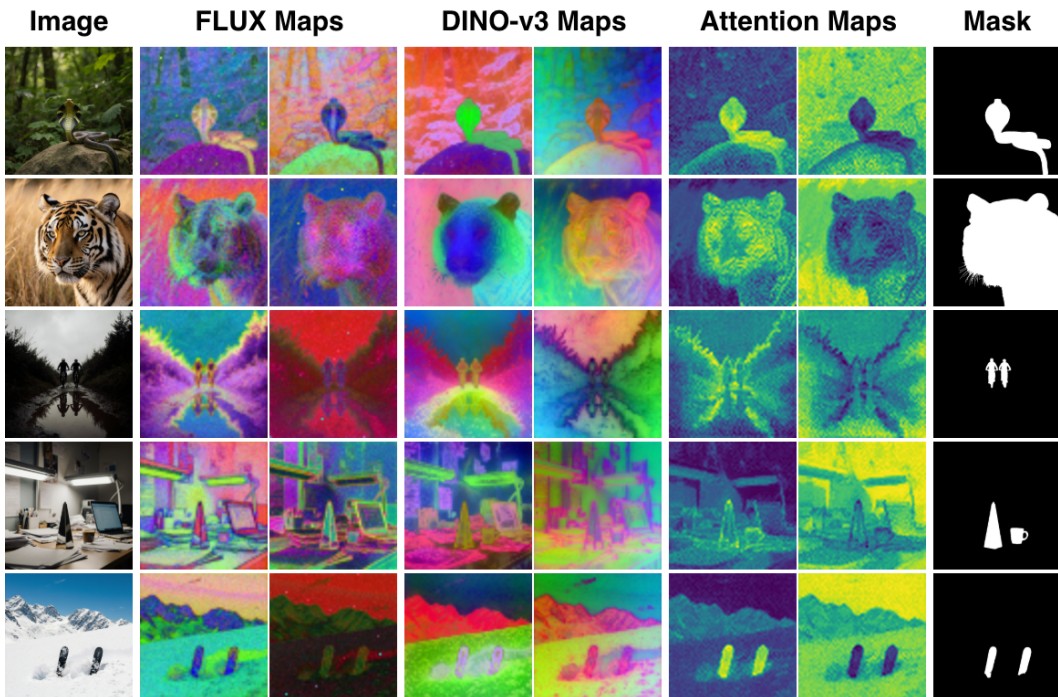

Figure 6: **Multi-Modal Feature Visualization:** Each modality captures complementary information: concept attention maps provide semantic localization, DINO-v3 encodes fine-grained visual semantics, and FLUX DiT features capture spatial scene structure. High-dimensional features (FLUX, DINO-v3) are visualized via PCA projection to RGB.

## B   PROMPTING

To further enhance the quality and diversity of our synthetic data, we employed an LLM (Achiam et al., 2023) to generate detailed, specific prompts rather than using simple class names. The prompting strategy was designed to systematically vary key aspects of scene composition, including object size, positioning, occlusion levels, lighting conditions, and environmental complexity. For example, when generating "lion" category images, prompts varied from scenes with multiple lions to single lions in challenging environmental conditions. These detailed textual descriptions guided the diffusion model to create more challenging, diverse training samples that better reflect real-world scenarios and edge cases. The set of example prompts for the "lion" category includes:

1. A medium-sized lion lying on a sunlit rock, partially obscured by tall grass, with a dense forest background; intricate shadows play on the lion's fur and the rock surface.

2. A small lion cub, occupying the left third of the frame, peeking through a thicket of dry branches in a savannah setting with blurred golden grass and a distant treeline.

3. Two lions resting under the shade of an acacia tree, one lion partially hidden by the tree's trunk; dappled sunlight filters through the leaves, creating complex patterns on the ground.

4. A majestic lion standing on a hilltop, backlit by the setting sun, casting a dramatic silhouette against a vibrant, cloud-streaked sky with the savannah stretching out in the background.

5. A close-up of a lion's face, centered in the frame, with its mane blending into a similarly colored rocky background; fine textures of the fur and rock are sharply defined.

6. A trio of lions walking through a misty grassland, with their figures partly obscured by the fog; subtle variations in coloration and mane distinguish each lion.

7. A lioness crouching low in a field of tall yellow grass, partially obscured and camouflaged by the foliage, with a clear blue sky above and distant hills in the background.

The full system prompt is presented in Figure 11

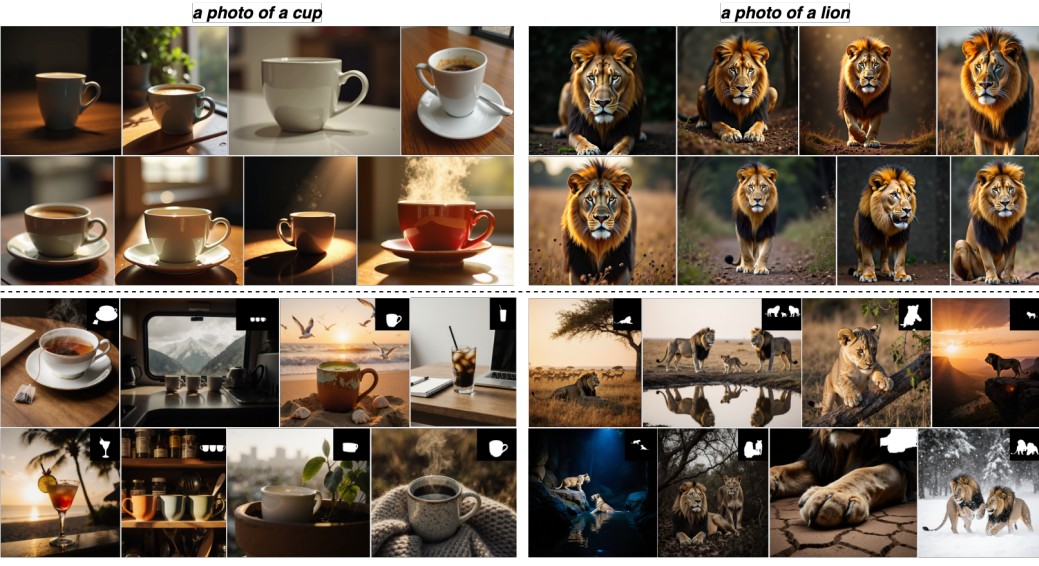

Figure 7: **Prompt Enhancement:** Top: Class name as a prompt. Bottom: LLM Prompt Generator. By focusing on key properties of salient object detection dataset the agent creates detailed and diverse prompts to maximize the diversity and realism.

## C DATASET QUALITY

Table 11: **Dataset Quality Comparison:** S3OD generated with large DiT model fine-tuned for photorealism achieves substantially higher quality and better real-data alignment compared to existing synthetic approaches, demonstrating the importance of realistic generation models.

| Method | Diffusion Model | Inception Score ↑ | FID ↓ |
|---|---|---|---|
| S3OD | FLUX-Krea (Black Forest Labs, 2025) | **35.19** | **1.74** |
| S3OD | FLUX-dev (Labs, 2023) | 31.94 | 1.90 |
| MaskFactory | Stable Diffusion (Rombach et al., 2022) | 17.41 | 2.81 |
| DatasetDM | Stable Diffusion (Rombach et al., 2022) | 14.97 | 3.16 |

We conducted a quality assessment of our synthetic dataset across multiple dimensions. Manual verification of 1,000 randomly sampled masks revealed high annotation quality: only 14 samples (1.4%) exhibited minor issues such as slightly incomplete mask boundaries, while merely 1 sample

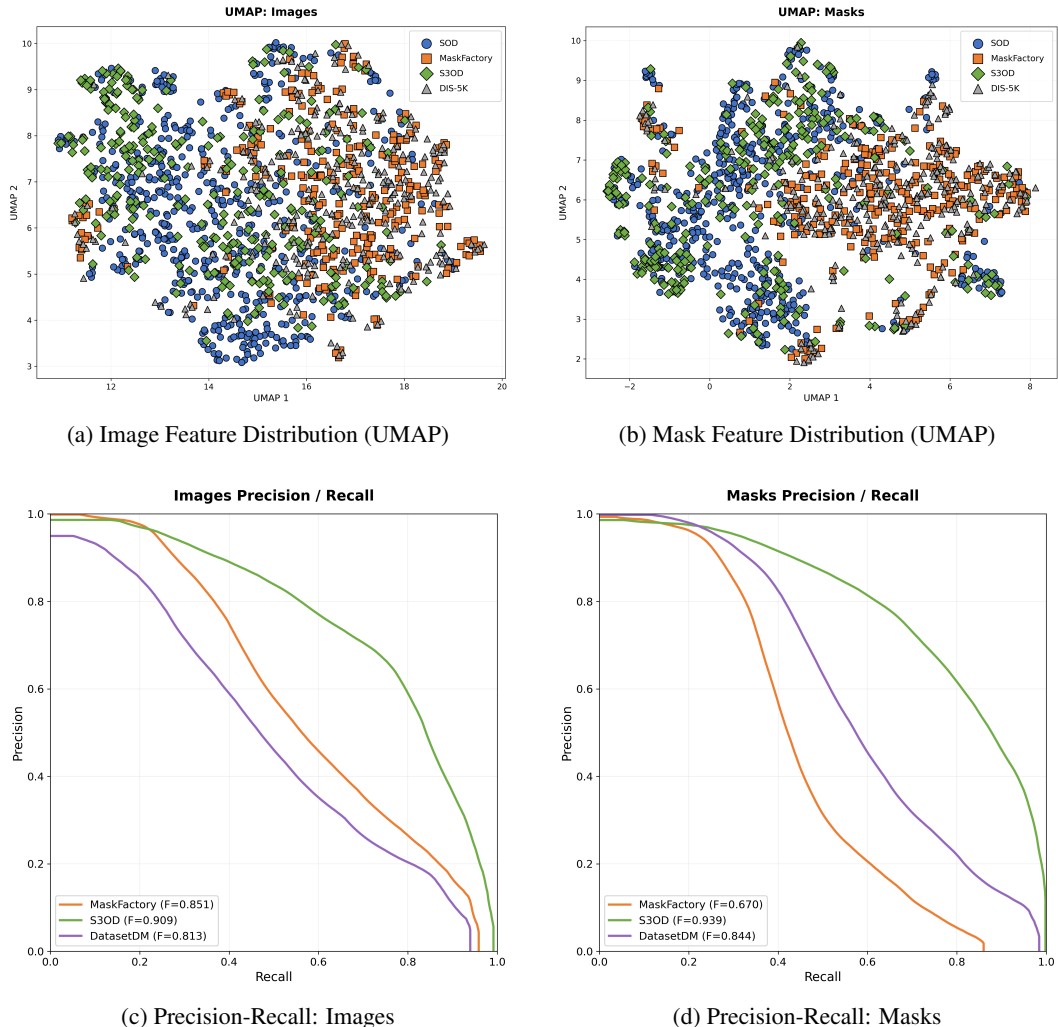

(a) Image Feature Distribution (UMAP)     (b) Mask Feature Distribution (UMAP)

(c) Precision-Recall: Images     (d) Precision-Recall: Masks

Figure 8: **Domain Gap Analysis.** (a-b) UMAP projections of DINO-v3 image and masks features. S3OD covers large portion of the real data distribution matching combined SOD real datasets. (c-d) Precision-Recall curves (Kynkäänniemi et al., 2019) vs a combination of DIS and SOD dataset: S3OD achieves higher recall and precision for both images and masks, compared to other synthetic datasets that only cover a part of the real data distribution, demonstrating a lower synthetic to real gap.

(0.1%) was missing a clear foreground object entirely. This demonstrates the effectiveness of our multi-stage filtering pipeline and multi-modal dataset diffusion approach.

To quantitatively evaluate synthetic-to-real domain gap we compute quality and coverage of the samples produced by a generative model following (Kynkäänniemi et al., 2019) versus a combination of SOD and DIS datasets. We observe that both synthetic images and masks closely follow real distribution in contrast to other methods that only model a part of it. Further, UMAP (McInnes et al., 2018) projections of DINO-v3 image and masks features demonstrate that S3OD samples cover a larger region of the real data manifold compared to MaskFactory (Qian et al., 2024). Reduced domain gap directly explains the superior generalization of the models trained on S3OD.

Another significant challenge in synthetic data generation is the domain gap between synthetic and real images. We observed that standard FLUX model fine-tuned for aesthetics produce unnaturally oversaturated images that differ substantially from real-world photography. To address this, we employ the FLUX-Krea checkpoint (Black Forest Labs, 2025), which underwent large-scale reinforcement learning alignment specifically for photorealism, producing significantly more natural-

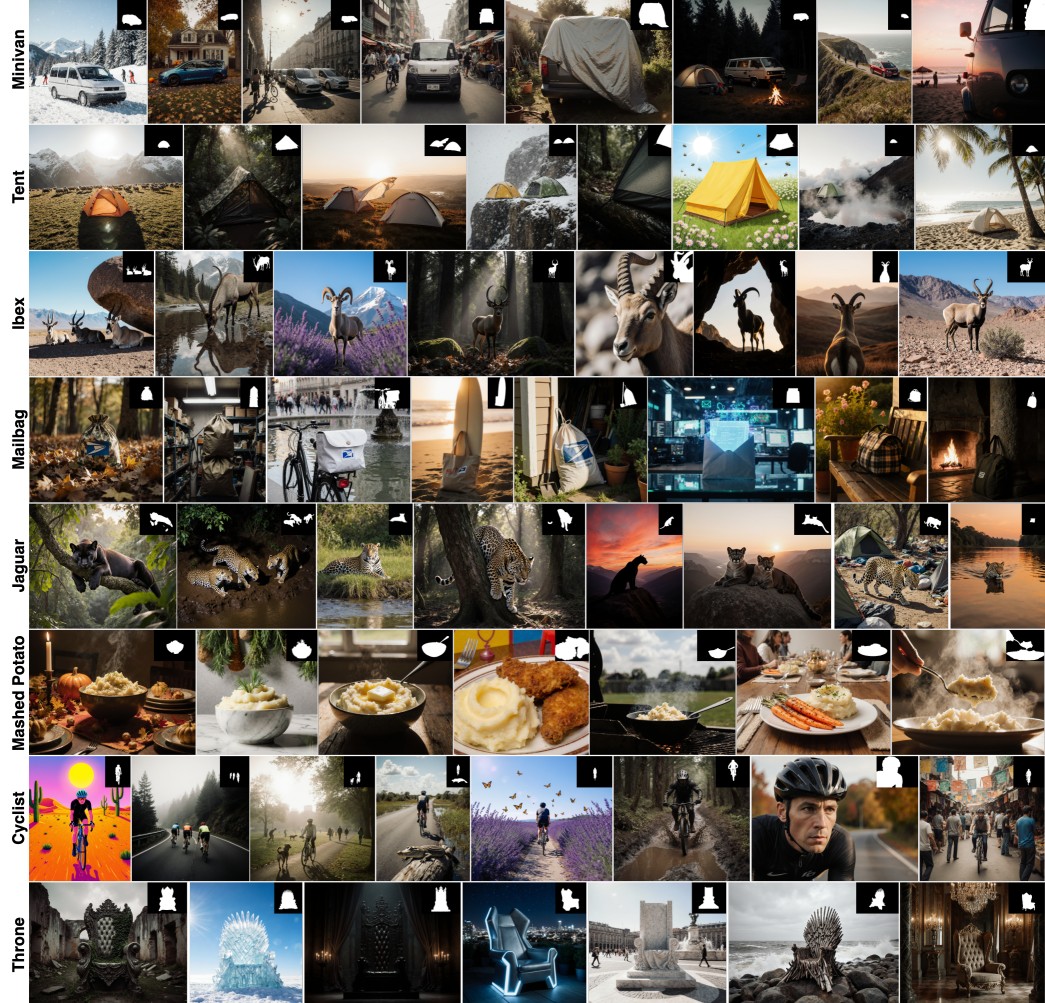

Figure 9: **S3OD Dataset Samples:** Our method generates diverse high-quality samples across a wide variety of object categories.

looking images. Additionally, during pretraining we apply comprehensive image augmentations to further reduce the synthetic-to-real domain gap.

We evaluate synthetic data quality using standard generative model metrics compared to existing approaches. As shown in Table 11, our method achieves superior image quality and diversity compared to datasets based on older diffusion models. S3OD achieves an Inception Score of 35.19 compared to MaskFactory's 17.41 and DatasetDM's 14.97, indicating better diversity and quality. Our FID score of 1.74 significantly outperforms MaskFactory (2.81) and DatasetDM (3.16), demonstrating closer similarity to real data distribution.

# D    QUALITATIVE EVALUATION

We visualize the random samples from different categories of S3OD in Figure 9. It demonstrates the diversity and realism achieved by our synthetic data generation pipeline, spanning various object types, lighting conditions, and scene compositions. The samples exhibit challenging scenarios with complex backgrounds, partial occlusions, and varying object: key attributes for training robust salient object detection models. As shown in Figure 7, LLM-based prompt generation significantly enhances the visual quality and diversity.

## E    MODEL DETAILS

S3ODNet achieves a strong balance between performance and efficiency as shown in Table 12, comparable to other state-of-the-art models that utilize large transformer backbones. Notably, the model is both more efficient and has more parameters compared to models that are based on the Swin architecture (Liu et al., 2021b). The DINO-v3 (Siméoni et al., 2025) backbone with ViT-B (Dosovitskiy et al., 2020) offers a favorable trade-off between computational efficiency and state-of-the-art performance.

Table 12: **Model Efficiency.** S3ODNet achieves comparable performance to other state-of-the-art salient object detection methods.

| Model | Total Parameters | FLOPs (T) | FPS |
|---|---|---|---|
| InSPyreNet (Kim et al., 2022) | 90,721,443 | 1.495 | 2.88 |
| BiRefNet (Zheng et al., 2024) | 220,176,498 | 1.143 | 3.65 |
| MVANet (Yu et al., 2024) | 94,139,021 | 0.857 | 4.62 |
| S3ODNet | 116,905,286 | 0.807 | 3.80 |

## F    STATE-OF-THE-ART COMPARISON

| Input Image | InSPyReNet | BiRefNet | MVANet | S3OD | Ground Truth |
|---|---|---|---|---|---|

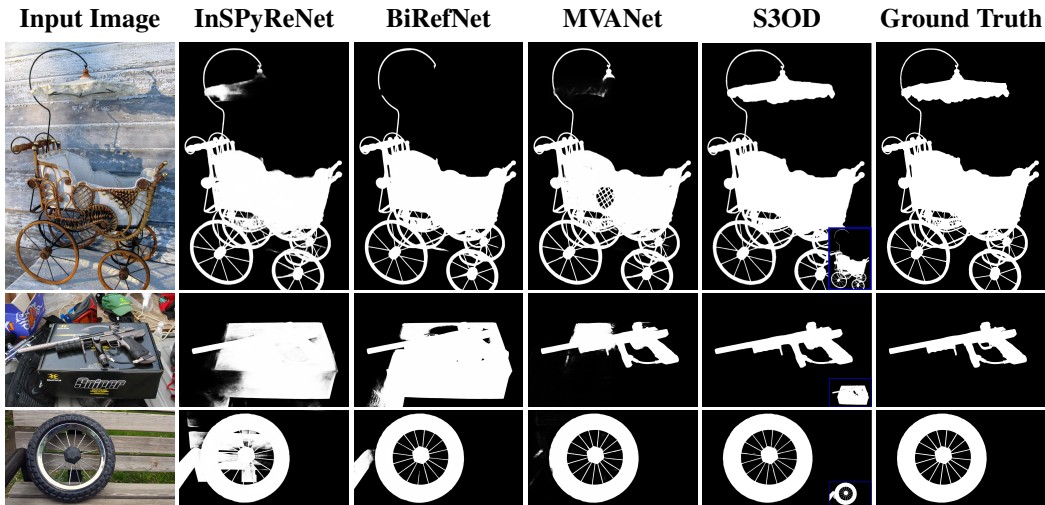

Figure 10: **Qualitative Comparison:** We compare S3ODNet vs state-of-the-art methods on DIS-5K (Qin et al., 2022) dataset. By modeling multiple hypothesis S3ODNet is able to predict detailed masks with high confidence. Alternative prediction can be seen in the bottom right corner.

We further expand the analysis of S3OD performance vs other state-of-the-art methods. Table 13 evaluates the performance compared to models finetuned from the foundational segmentation model (Ravi et al., 2024). We observe that all models that are based on SAM perform well on simpler subset of DIS (DIS-TE1) but the performance drops significantly as the sample complexity increases. S3ODNet outperforms all approaches (Ke et al., 2023; Liu et al., 2024) matching the performance of DIS-SAM (Liu et al., 2025), which is a more complex two-stage pipeline consisting of two separate models performing segmentation in high resolution, resulting in significantly larger complexity and number of parameters compared to our simple network design. This evaluation demonstrates that the limited manually labeled data is still insufficient to finetune even the state-of-the-art foundational models pretrained on various data from a slightly different domain.

Next we provide the results of more state-of-the-art methods as well as S3ODNet variant trained only on DIS-5K or SOD datasets in Table 14 to further evaluate the impact of pretraining on synthetic data. We include (Wei et al., 2020a; Zeng et al., 2019; Tang et al., 2021; Xie et al., 2022a) model to SOD evaluation. Interestingly, S3OD trained only on synthetic data outperforms most of the older methods that were trained on SOD datasets when evaluating on SOD benchmarks! This showcases both the quality of the synthetic data and model effectiveness. S3ODNet trained only on SOD

confirms the insights from Section 5 – the performance on salient object detection benchmarks is saturated. All transformer-based methods that were trained on SOD data show comparable performance when evaluating on the same datasets. The only benchmark that is from a different data distribution is DUT-OMRON, demonstrating that S3ODNet trained on SOD outperforms other methods and pretraining on S3OD further improves performance. This also highlights the importance of cross-dataset generalization evaluation instead of only measuring overfitting to small academic benchmarks.

The evaluation of S3ODNet trained on DIS-5K follows the same trend. We further evaluate (Qin et al., 2019; 2020; Xie et al., 2022a; Qin et al., 2022; Pei et al., 2023; Zhou et al., 2023). Similarly to other evaluations, S3ODNet trained on DIS outperforms other methods trained on same dataset and pretraining on S3OD further improves the performance.

Table 13: **Comparison of SAM-based methods and S3ODNet:** Our model outperforms most larger models finetuned from Segment Anything matching the performance of complex two-stage pipeline (Liu et al., 2025).

| Method | DIS-TE1 | | | | DIS-TE2 | | | | DIS-TE3 | | | | DIS-TE4 | | | | Overall | | | |
|---|---|---|---|---|---|---|---|---|---|---|---|---|---|---|---|---|---|---|---|---|
| | $F_m \uparrow$ | $S_\alpha \uparrow$ | $E_M^\Phi \uparrow$ | MAE↓ | $F_m \uparrow$ | $S_\alpha \uparrow$ | $E_M^\Phi \uparrow$ | MAE↓ | $F_m \uparrow$ | $S_\alpha \uparrow$ | $E_M^\Phi \uparrow$ | MAE↓ | $F_m \uparrow$ | $S_\alpha \uparrow$ | $E_M^\Phi \uparrow$ | MAE↓ | $F_m \uparrow$ | $S_\alpha \uparrow$ | $E_M^\Phi \uparrow$ | MAE↓ |
| SAM | .838 | .843 | .805 | .047 | .803 | .792 | .863 | .081 | .773 | .761 | .848 | .094 | .677 | .697 | .762 | .162 | .773 | .773 | .845 | .096 |
| HQ-SAM | .903 | .907 | .959 | .019 | .895 | .883 | .950 | .029 | .860 | .851 | .926 | .045 | .776 | .799 | .863 | .088 | .859 | .860 | .924 | .045 |
| Pi-SAM | .890 | .894 | .947 | .027 | .903 | .907 | .953 | .027 | .899 | .901 | .953 | .030 | .869 | .871 | .939 | .046 | .890 | .893 | .948 | .033 |
| DIS-SAM | **.929** | **.929** | **.960** | **.019** | **.924** | .921 | **.955** | **.025** | .918 | .908 | .948 | .030 | .899 | .888 | .932 | .043 | **.917** | **.911** | .949 | **.029** |
| S3ODNet | .892 | .902 | .932 | .031 | .923 | **.921** | .953 | .026 | **.930** | **.920** | **.960** | **.025** | **.909** | **.902** | **.954** | **.034** | .914 | **.911** | **.950** | **.029** |

Table 14: **Quantitative Comparison:** We extend the comparison to more baselines and also evaluate S3ODNet trained only on real data. S3ODNet trained on the same datasets as prior work demonstrates better performance. Pretraining on S3OD further improve the performance, showing the value of the dataset even on saturated benchmarks.

| Method | Data | DAVIS-S | | | | HRSOD-TE | | | | UHRSD-TE | | | | DUTS-TE | | | | DUT-OMRON | | | |
|---|---|---|---|---|---|---|---|---|---|---|---|---|---|---|---|---|---|---|---|---|---|
| | | $F_m \uparrow$ | $S_\alpha \uparrow$ | $E_M^\Phi \uparrow$ | MAE↓ | $F_m \uparrow$ | $S_\alpha \uparrow$ | $E_M^\Phi \uparrow$ | MAE↓ | $F_m \uparrow$ | $S_\alpha \uparrow$ | $E_M^\Phi \uparrow$ | MAE↓ | $F_m \uparrow$ | $S_\alpha \uparrow$ | $E_M^\Phi \uparrow$ | MAE↓ | $F_m \uparrow$ | $S_\alpha \uparrow$ | $E_M^\Phi \uparrow$ | MAE↓ |
| LDF | SOD | .911 | .922 | .947 | .019 | .904 | .904 | .919 | .032 | .888 | .913 | .891 | .047 | .892 | .898 | .910 | .034 | .820 | .838 | .873 | .051 |
| HRSOD | SOD | .899 | .876 | .955 | .026 | .905 | .896 | .934 | .030 | - | - | - | - | .835 | .824 | .885 | .050 | .743 | .762 | .831 | .065 |
| DHQ | SOD | .938 | .920 | .947 | .012 | .922 | .920 | .947 | .022 | .900 | .911 | .905 | .039 | .894 | .900 | .919 | .031 | .820 | .836 | .873 | .045 |
| PGNet | SOD | .957 | .954 | .979 | .010 | .945 | .938 | .946 | .020 | .935 | .949 | .916 | .020 | .859 | .871 | .897 | .038 | .772 | .786 | .884 | .058 |
| InSpyreNet | SOD | .977 | .973 | .987 | .007 | .956 | .956 | .962 | .018 | .957 | .953 | .965 | .020 | .932 | .936 | .956 | .024 | .823 | .872 | .906 | .046 |
| BiRefNet | SOD | **.979** | **.975** | .989 | .006 | **.963** | .957 | .973 | .016 | .963 | **.957** | **.969** | .016 | .943 | .944 | .962 | .018 | .839 | .882 | .896 | .038 |
| S3ODNet | SOD | .975 | .969 | .991 | .005 | .964 | .953 | .973 | .017 | .964 | .948 | .967 | .019 | .951 | .939 | .966 | .018 | .874 | .890 | .919 | .033 |
| S3ODNet | S3OD + SOD | **.979** | .974 | **.993** | **.004** | .963 | **.961** | **.978** | **.013** | **.964** | .952 | **.969** | .018 | **.954** | **.949** | **.972** | **.015** | **.879** | **.898** | **.924** | **.032** |

| Method | Data | DIS-1 | | | | DIS-2 | | | | DIS-3 | | | | DIS-4 | | | | Overall | | | |
|---|---|---|---|---|---|---|---|---|---|---|---|---|---|---|---|---|---|---|---|---|---|
| | | $F_m \uparrow$ | $S_\alpha \uparrow$ | $E_M^\Phi \uparrow$ | MAE↓ | $F_m \uparrow$ | $S_\alpha \uparrow$ | $E_M^\Phi \uparrow$ | MAE↓ | $F_m \uparrow$ | $S_\alpha \uparrow$ | $E_M^\Phi \uparrow$ | MAE↓ | $F_m \uparrow$ | $S_\alpha \uparrow$ | $E_M^\Phi \uparrow$ | MAE↓ | $F_m \uparrow$ | $S_\alpha \uparrow$ | $E_M^\Phi \uparrow$ | MAE↓ |
| BASNet | DIS | .663 | .741 | .756 | .105 | .738 | .781 | .808 | .096 | .790 | .816 | .848 | .080 | .785 | .806 | .844 | .087 | .744 | .786 | .814 | .092 |
| U$^2$Net | DIS | .701 | .762 | .783 | .085 | .768 | .798 | .825 | .083 | .813 | .823 | .856 | .073 | .800 | .814 | .837 | .085 | .771 | .799 | .825 | .082 |
| PGNet | DIS | .754 | .800 | .848 | .067 | .807 | .833 | .880 | .065 | .843 | .844 | .911 | .056 | .831 | .841 | .899 | .065 | .809 | .830 | .885 | .063 |
| IS-Net | DIS | .740 | .787 | .820 | .074 | .799 | .823 | .858 | .070 | .830 | .836 | .883 | .064 | .827 | .830 | .870 | .072 | .799 | .819 | .858 | .070 |
| FP-DIS | DIS | .784 | .821 | .860 | .060 | .827 | .845 | .893 | .059 | .868 | .871 | .922 | .049 | .846 | .852 | .906 | .061 | .831 | .847 | .895 | .047 |
| UDUN | DIS | .784 | .817 | .864 | .059 | .829 | .843 | .886 | .058 | .865 | .865 | .917 | .050 | .846 | .849 | .901 | .059 | .831 | .844 | .892 | .057 |
| SAM-HQ | DIS | **.897** | **.907** | **.943** | **.019** | .889 | .883 | .928 | .029 | .851 | .851 | .897 | .045 | .763 | .799 | .843 | .088 | .850 | .860 | .903 | .045 |
| InSpyreNet | DIS | .845 | .873 | .874 | .043 | .894 | .905 | .916 | .036 | .919 | .918 | .940 | .034 | .905 | **.905** | .936 | .042 | .891 | .900 | .917 | .039 |
| BiRefNet | DIS | .860 | .885 | .911 | .037 | .894 | .900 | .930 | .036 | .925 | .919 | .955 | .028 | .904 | .900 | .939 | .039 | .896 | .901 | .934 | .035 |
| MVANet | DIS | .862 | .880 | .906 | .039 | .909 | .912 | .942 | .032 | .924 | .918 | .954 | .030 | .907 | **.905** | .946 | .039 | .900 | .904 | .937 | .035 |
| S3ODNet | DIS | .896 | .891 | .928 | .031 | .919 | .905 | .943 | .030 | .928 | .910 | .957 | .028 | .896 | .883 | .942 | .039 | .910 | .897 | .943 | .032 |
| S3ODNet | DIS + S3OD | .892 | .902 | .932 | .031 | **.923** | **.921** | **.953** | **.026** | **.930** | **.920** | **.960** | **.025** | **.909** | .902 | **.954** | **.034** | **.914** | **.911** | **.950** | **.029** |

## G  MULTI-MASK DECODER ANALYSIS

Our multi-mask decoder builds upon the multiple hypothesis prediction (MHP) framework of (Rupprecht et al., 2017), which shows that predicting $M$ hypotheses creates a Voronoi tessellation of the output space, with each hypothesis converging to the conditional mean of its region. However, salient object detection differs fundamentally from inherently ambiguous tasks like future prediction: most samples have a single clear ground truth and only a small subset are truly ambiguous (multiple objects or complex scene). This creates a critical training instability. Without explicit regularization, branches that are initially far from the data receive no gradients from the best-match selection $i^* = \arg\min_i \text{IoU}(m_i, y)$ and degenerate, as most samples assign to a single dominant branch. This is why we introduce auxiliary loss with exponential decay $L = L_{i^*} + \lambda_{reg} e^{-\gamma t} \sum_i L_i$, which prevents branch collapse by forcing all branches to maintain proximity to ground truth early in training, then gradually allows diverse outputs as the decay reduces supervision. This setup enables branches to handle both the dominant unambiguous cases and the sparse ambiguous samples. The ablation study below validates this design. The baseline configuration achieves an optimal balance

between branch diversity and segmentation performance. Without auxiliary loss, we observe branch collapse as two branches stop receiving gradients and output empty masks. Static regularization without decay produces overfits to output all similar masks ignoring the ambiguity, while stronger regularization or slower decay both slightly reduce entropy without clear performance benefits.

Table 15: **Multi-Mask Decoder Loss Ablation:** We report segmentation performance on UHRSD-TE and DUT-OMRON benchmarks, along with diversity metrics computed across all test samples.

| $\lambda_{reg}$ | $\gamma$ | Diversity Metrics | | UHRSD-TE | | | | DUT-OMRON | | | |
|---|---|---|---|---|---|---|---|---|---|---|---|
| | | Entropy↑ | Avg IoU↓ | $F_m \uparrow$ | $S_\alpha \uparrow$ | $E_M^\Phi \uparrow$ | MAE↓ | $F_m \uparrow$ | $S_\alpha \uparrow$ | $E_M^\Phi \uparrow$ | MAE↓ |
| 0.1 | 0.2 | .878 | .863 | .964 | .948 | .967 | .019 | .874 | .890 | .919 | .033 |
| 0.2 | 0.2 | .823 | .869 | .963 | .948 | .967 | .020 | .873 | .891 | .917 | .033 |
| 0.1 | 0.1 | .824 | .877 | .962 | .948 | .967 | .020 | .873 | .890 | .916 | .034 |
| 0.1 | 0.0 | .906 | .945 | .962 | .949 | .968 | .019 | .874 | .890 | .919 | .034 |
| 0.0 | 0.0 | 0.0 | 0.0 | .964 | .947 | .966 | .020 | .876 | .890 | .920 | .034 |

We evaluate the impact of auxiliary branch regularization through the $\lambda_{reg}$ and decay rate $\gamma$ parameters in our multi-mask decoder loss formulation. The baseline configuration uses $\lambda_{reg} = 0.1$ with exponential decay $\gamma = 0.2$.

Due to the computational cost of retraining the model, we cannot perform an exhaustive grid search over all possible parameter combinations. Instead, we strategically select four key ablation variants that test fundamental design choices: (1) stronger regularization ($\lambda_{reg} = 0.2$) to assess if auxiliary branches benefit from full mask supervision, (2) slower decay ($\gamma = 0.1$) to maintain full mask longer during training, (3) static regularization ($\gamma = 0.0$) without any decay to evaluate the necessity of the temporal annealing mechanism, and (4) no auxiliary loss ($\lambda_{reg} = 0.0$) training only the best-matching branch to test if some branches stop receiving gradient during the training.

These variants assess the trade-off between enforcing branch diversity and preventing degradation of unused predictions. The last configuration ($\lambda_{reg} = 0.0$) is particularly important as it tests whether supervising all branches with the ground-truth mask in early epochs provides any benefit and stabilizes the training.

## H   LIMITATIONS AND BROADER IMPACT

S3OD data is fully generated, so we deliberately don't provide a test split for the dataset, as we believe the methods can be pretrained on synthetic data, but should be evaluated on smaller-scale, precise human annotations. The multi-stage filtering strategy detects and removes most of the fail cases but the model occasionally might produce some artifacts both while generating an image or mask, such as mask not fully covering an object or a scene missing a clear salient object. We acknowledge the high computational cost of generating large-scale data using diffusion transformers, yet the process is still orders of magnitude faster than manual labeling and can be effectively parallelized. Additionally, similarly to (Zheng et al., 2024) we observe that training for more than 100 epochs almost does not impact the metrics but slightly improves finer details quality so we were able to obtain similar metrics with using only 4 A6000 GPUs for 2.5 days which makes the training pipeline more accessible. We expect that the insights into the combination of generative and discriminative features, as well as the iterative data generation, can be reused in other tasks and domains, especially where obtaining the ground truth data is the main bottleneck for scaling.

## I   THE USE OF LARGE LANGUAGE MODELS (LLMS)

The LLM is a core part of the dataset generation method Figure 3 ensuring we build a large library of diverse captions for various object categories. We also used LLMs to polish the writing, verify grammar or improve the sentence structure.

**System Prompt for Salient Object Detection Data Generation**

Generate exactly {num_prompts} diverse, photorealistic prompts for {main_class} images for salient object detection. Create natural scenes with varying complexity levels.

**Requirements:**

- Photorealistic scenes only—no artistic or cartoon styles
- Main object clearly visible and identifiable
- Sharp focus throughout the scene
- Natural lighting and environments

**Vary these aspects across prompts:**

- **Object variations:** Different sizes, positions, quantities (1-3 objects), conditions (new/worn), and orientations
- **Scene complexity:** Mix simple and complex backgrounds—from clean settings to cluttered environments with multiple objects and busy textures
- **Lighting conditions:** Natural daylight, golden hour, overcast skies, indoor lighting, shadows and highlights
- **Environments:** Indoor/outdoor settings, natural habitats, functional contexts (object being used), storage areas, seasonal variations
- **Visual challenges:** Include some scenes with partial occlusion, similar colors, reflective surfaces, overlapping objects, or camouflage effects when natural
- **Perspectives:** Close-ups, medium shots, wide views, and varied camera angles (above, below, side views)
- **Context diversity:** Objects in use, at rest, in groups, in natural habitats, different weather conditions, times of day

**Balance:** Create a mix of challenging segmentation scenarios.

Return exactly {num_prompts} prompts as Python list: `[''A scene description...'', ...]`

**Important:** Maximize diversity—avoid repetitive scenarios or settings.

Figure 11: The complete system prompt used to instruct the LLM (Achiam et al., 2023) for generating diverse text descriptions. These descriptions create photorealistic scenes with varying complexity, object configurations, and environmental conditions to simulate challenging real-world scenarios for salient object detection. The prompt emphasizes natural lighting, sharp focus, and balanced diversity across simple and complex backgrounds.

