# OpenReview forum: "S3OD: Towards Generalizable Salient Object Detection with Synthetic Data"
_ICLR.cc/2026/Conference — ICLR 2026 Poster_

### Official Review · Reviewer_n7Sk · 2025-10-27

**Soundness:** 3
**Presentation:** 3
**Contribution:** 3
**Rating:** 6
**Confidence:** 4

**Summary:**

This paper introduces S3OD, a large-scale synthetic dataset (139k images) and a corresponding ambiguity-aware salient object detection (SOD) model. The work proposes a multi-modal diffusion pipeline that fuses FLUX DiT features, concept attention maps, and DINO-v3 embeddings to generate high-quality image–mask pairs. An iterative generation framework prioritizes difficult categories based on model feedback, and a multi-mask decoder handles label ambiguity by predicting multiple plausible saliency maps. Experiments on DIS-5K, HRSOD, UHRSD, and DUT-OMRON show strong cross-dataset generalization, with models trained purely on synthetic data approaching or surpassing state-of-the-art methods.

**Strengths:**

1.Novel synthetic-data pipeline that unifies diffusion- and representation-based cues (DiT + DINO-v3 + concept maps), addressing weaknesses of prior attention-extraction methods.

2.Large-scale dataset contribution with detailed filtering and iterative feedback, potentially valuable to the SOD community.

3.Ambiguity-aware decoding design that explicitly models multiple valid masks, a practically important perspective for SOD.

**Weaknesses:**

1.While the dataset generation process is well described, the core algorithmic novelty of the ambiguity-aware model is incremental—largely a DPT backbone with multi-branch outputs and IoU-guided selection. This architectural contribution may not reach ICLR’s expected level of theoretical innovation.
2.The evaluation relies mainly on FID and Inception Score; no user study or domain-gap quantification is provided to show the practical relevance of the generated images for real-world deployment.
3.The comparison with other methods may not control for data volume and pretraining effects (e.g., DINO-v3 backbone vs. Swin-B), so performance gains might arise from model capacity rather than the proposed framework.
4.Critical hyperparameters (diffusion steps, fusion architecture, loss weights) are partially specified but not fully justified. The claim of “2 days on 8×H200” suggests heavy computation that may not be accessible to the community.

**Questions:**

1.The treatment of ambiguity remains heuristic. The paper models uncertainty through multiple mask branches and a decaying regularizer but lacks a probabilistic or information-theoretic formulation.
2.Loss functions (Focal + IoU) and weighting constants (e.g., λ_mask = 10, γ = 0.2) are empirically set without analytical justification or stability analysis.
3.The relaxed branch assignment mitigates degeneration empirically but offers no convergence proof or theoretical characterization of branch usage.
4.Domain-gap evaluation between synthetic and real data relies solely on FID and Inception Scores—useful but insufficient for capturing distributional or representational discrepancies.
5.The experimental section is extensive and well organized: the authors evaluate cross-dataset generalization, multi-modal ablations, multi-round data synthesis, and backbone comparisons. This breadth supports the main empirical claim that synthetic data can significantly improve SOD generalization. Nevertheless, several confounding factors remain unresolved. The performance gains might partially stem from larger data volumes and stronger pretrained encoders (DINO-v3) rather than from the proposed mechanisms themselves. Moreover, reproducibility is limited by high computational cost (three data-generation rounds, filtering, and multi-GPU training) and incomplete sensitivity analyses of key hyperparameters.

---

> ### Author Response · Authors · 2025-11-21
> **Official Comment by Authors**
>
> Thank you for the thoughtful and positive review of our work! Below we address main questions and concerns:
>
> > **The core algorithmic novelty of the ambiguity-aware model is incremental---largely a DPT backbone with multi-branch outputs and IoU-guided selection. This architectural contribution may not reach ICLR's expected level of theoretical innovation.**
>
> We disagree with the reviewer that a paper's novelty solely depends on a novel network architecture. The vast majority of ICLR papers do not contribute new architectures.
>
> The novelty of our paper lies in
> 1) Demonstrating a data generation pipeline that can improve tasks with limited annotated data, especially for generalization.
> 2) Introducing an ambiguity-aware model (S3ODNet) to the SOD task, which has not been considered before.
> 3) Releasing a large-scale novel, synthetic dataset for this task to allow for future improvements, now that large-scale training is a possibility.
>
> > **The evaluation relies mainly on FID and Inception Score; no user study or domain-gap quantification is provided to show the practical relevance of the generated images for real-world deployment.**
>
> Thanks, we agree that additional domain gap metrics would further strengthen the analysis of the dataset quality. To quantitatively evaluate the synthetic-to-real domain gap, we compute the quality and coverage of the samples produced by a generative model versus a combination of SOD and DIS datasets. We observe that both synthetic images and masks closely follow real distribution in contrast to other methods that only model a part of it. Further, UMAP projections of DINO-v3 image and masks features demonstrate that S3OD samples cover a larger region of the real data manifold compared to MaskFactory . The reduced domain gap directly explains the superior generalization of models trained on S3OD. We have added the results to Figure 11 in Appendix E.
>
> > **The comparison with other methods may not control for data volume and pretraining effects (e.g., DINO-v3 backbone vs. Swin-B), so performance gains might arise from model capacity rather than the proposed framework.**
>
> Table 2 and Table 7 control for this difference. Additionally, we train prior work on our dataset. With a fixed architecture, our dataset improves the performance. With a fixed dataset, our model shows better performance. This validates both components. For further verification that the dataset is valuable not only to our method, we train prior work using our dataset and show that this also yields consistent improvements. We have added the extra experiment to Table 9 in the revised manuscript.
>
> > **Critical hyperparameters (diffusion steps, fusion architecture, loss weights) are partially specified but not fully justified. The claim of "2 days on 8×H200" suggests heavy computation that may not be accessible to the community.**
>
> - Generation: We follow the standard DiT generation setup without changing base hyperparameters: 28 timesteps, 1024p resolution, etc.
> - Loss setup matches Segment Anything, a common choice for training a segmentation model.
> - Fusion Architecture: We design a simple, intuitive fusion block: The 256-dimensional projection space is chosen to match the original DPT decoder implementation, ensuring computational parity with baseline methods (Table 10).
>
> Similar to BiRefNet, we observe that training for a large number of epochs almost does not impact the metrics while slightly improving finer details segmentation. We achieved comparable results with 2 days of training on 4 A6000 GPUs.
>
> > **The treatment of ambiguity remains heuristic. The paper models uncertainty through multiple mask branches and a decaying regularizer, but lacks a probabilistic or information-theoretic formulation.**
>
> A detailed theoretical analysis can be found in Rupprecht et al. (2017). In short, the formulation learns to approximate the conditional distribution of masks with $N$ point samples. The regularizer prevents branches from dying off early in the training in case they are never selected, but it is not needed close to convergence. If the reviewer thinks it is helpful, we can also include a short discussion in our paper.

---

> ### Author Response · Authors · 2025-11-21
> **Official Comment By Authors**
>
> > **The relaxed branch assignment mitigates degeneration empirically but offers no convergence proof or theoretical characterization of branch usage.**
>
> The model approximates the target distribution through $N$ point estimates, effectively dividing the output space into $N$ Voronoi regions $R_n$ induced by the branches. At convergence, the expected error $\int_R p(x)\mathcal{L}(x)dx$ in each region will be equal. More details can be found in Rupprecht et al. (2017).
>
> > **The performance gains might partially stem from larger data volumes and stronger pretrained encoders (DINO-v3) rather than from the proposed mechanisms themselves. Moreover, reproducibility is limited by high computational cost (three data-generation rounds, filtering, and multi-GPU training) and incomplete sensitivity analyses of key hyperparameters.**
>
> Yes! All these factors indeed contribute to the overall performance as shown in Table 2, Table 4, Table 6, and Table 7. Enabling learning from larger data volumes due to generating synthetic data is one of the key contributions of our work. Nonetheless, the paper shows that each component individually improves the performance in a fair setting compared to prior work, which validates all our claims.
>
> We will release the complete code, pretrained models, and generated data. The generation process can be easily parallelized and processed in chunks, with all components runnable on consumer-grade GPUs (e.g., A6000).

---

> > ### Comment · Reviewer_n7Sk · 2025-11-27
> >
> > Thank you for the detailed and constructive response. The authors have successfully addressed several major concerns through supplemental experiments and clarifications. The addition of UMAP visualizations and Quality/Coverage metrics is a compelling way to strengthen the domain gap analysis, and the cross-validation experiments (Table 9) effectively disentangle the contributions of the new dataset and the S3ODNet model.
> >
> > However, a few critical points regarding theoretical grounding and reproducibility still require final clarification and must be incorporated into the revised manuscript:
> >
> > ### 1. Algorithmic Novelty and Theoretical Foundation
> >
> > I accept that the paper's primary contribution lies in the system-level innovation—the data generation pipeline, the large-scale dataset, and the application of an ambiguity-aware model to the SOD task. Regarding the S3ODNet model itself, citing Rupprecht et al. (2017) to ground the multi-branch approach theoretically is satisfactory and significantly boosts the model's theoretical credibility. The authors must incorporate this theoretical discussion and the corresponding citation into the main body or appendix of the revised paper. Furthermore, while the reference provides the theoretical basis, I still require the authors to explicitly articulate the unique algorithmic refinement or theoretical advantage of S3ODNet compared to the general multi-point estimation or multiple-choice learning frameworks presented in the cited work, to justify its specific algorithmic contribution.
> >
> > ### 2. Hyperparameter Justification and Sensitivity Analysis
> >
> > The explanation that many hyperparameters follow industry standards (DiT, DPT, SAM) is acceptable and mitigates the need for a full sensitivity analysis. However, the specific weights in the combined loss function (e.g., $\lambda_{mask}=10$ and $\lambda_{reg}=0.1$) are crucial for successful training, branch usage, and final performance. A sensitivity analysis for these specific loss weights is still needed. If high computational cost prevents a comprehensive analysis, the authors should clearly state this as a limitation in the manuscript and provide at least a minimal ablation study (e.g., varying the main weights by $\pm 50\%$) to demonstrate the stability of the chosen values.
> >
> > ### 3. Reproducibility and Computational Cost
> >
> > I appreciate the commitment to releasing the complete code, pretrained models, and generated data, as well as providing an alternative hardware setup using 4 A6000 GPUs. To fully resolve the reproducibility issue, the authors must explicitly disclose the specific training time required when using the 4 A6000 GPUs in the revised manuscript's experimental section, ensuring transparency for the community. Concurrently, the paper should frankly acknowledge the high computational cost associated with the multi-round data generation and multi-GPU training as a limitation that affects the ease of complete, from-scratch reproduction for most researchers.

---

> > > ### Author Response · Authors · 2025-12-03
> > >
> > > Thanks for your feedback! We are pleased to hear that several major concerns have been addressed. We agree that the proposed revisions will further strengthen the paper, have added these to the updated manuscript, and also provide a detailed answer below:
> > >
> > >
> > > > **1. Algorithmic Novelty and Theoretical Foundation**
> > >
> > > Thanks for the suggestion. We have expanded the analysis of the multi-mask-decoder in Appendix I. Salient object detection differs fundamentally from inherently ambiguous tasks like future prediction: most samples have a single clear ground truth, and only a small subset are truly ambiguous (multiple objects or complex scenes). This creates a critical training instability. Without explicit regularization, branches that are initially far from the data receive no gradients from the best-match selection $i^* = \arg\min_i \text{IoU}(m_i, y)$ and degenerate, as most samples are assigned to a single dominant branch. This is why we introduce auxiliary loss with exponential decay $L = L_{i^*} + \lambda_{reg}e^{-\gamma t}\sum_i L_i$, which prevents branch collapse by forcing all branches to maintain proximity to ground truth early in training, then gradually allows diverse outputs as the decay reduces supervision. This setup enables branches to handle both the dominant unambiguous cases and the sparse ambiguous samples.
> > >
> > >
> > > > **2. Hyperparameter Justification and Sensitivity Analysis**
> > >
> > > We do agree that an ablation study of combined loss parameters helps to better demonstrate the importance and sensitivity of the setup. We have added this experiment to Appendix I. The baseline configuration achieves an optimal balance between branch diversity and segmentation performance. Without auxiliary loss, we observe branch collapse as two branches stop receiving gradients and output empty masks. Static regularization without decay produces overfits to output all similar masks, ignoring the ambiguity, while stronger regularization or slower decay both slightly reduce entropy without clear performance benefits.
> > >
> > > | λ_reg | γ   | **Diversity** |          | **UHRSD-TE** |      |      |      | **DUT-OMRON** |      |      |      |
> > > |-------|-----|---------------|----------|--------------|------|------|------|---------------|------|------|------|
> > > |       |     | Entropy↑      | Avg IoU↓ | Fm↑          | Sm↑  | Em↑  | MAE↓ | Fm↑           | Sm↑  | Em↑  | MAE↓ |
> > > | 0.1   | 0.2 | .878          | .863     | .964         | .948 | .967 | .019 | .874          | .890 | .919 | .033 |
> > > | 0.2   | 0.2 | .823          | .869     | .963         | .948 | .967 | .020 | .873          | .891 | .917 | .033 |
> > > | 0.1   | 0.1 | .824          | .877     | .962         | .948 | .967 | .020 | .873          | .890 | .916 | .034 |
> > > | 0.1   | 0.0 | .906          | .945     | .962         | .949 | .968 | .019 | .874          | .890 | .919 | .034 |
> > > | 0.0   | 0.0 | 0.0           | 0.0      | .964         | .947 | .966 | .020 | .876          | .890 | .920 | .034 |
> > >
> > >
> > > > **3. Reproducibility and Computational Cost**
> > >
> > > Thanks for the suggestion. We do acknowledge the high computational cost of data generation, yet we also want to note that the generation pipeline is orders of magnitude faster than manual labelling 12sec vs. 10+min), so it is still more scalable than manually annotating images in high resolution. We have added this discussion to the Limitations section in Appendix J.

---

### Official Review · Reviewer_VcaT · 2025-11-01

**Soundness:** 2
**Presentation:** 2
**Contribution:** 2
**Rating:** 2
**Confidence:** 5

**Summary:**

In this paper, the authors proposed S3OD method for salient object detection. In addition, the authors introduced S3OD dataset. In particular, they have created a massive synthetic dataset of over 139000 high resolution images with machine generated labels. The S3OD dataset helps improve the performance of the S3OD method.

**Strengths:**

+ The authors introduced S3OD dataset with 139,000+ samples.
+ Models trained only on the S3OD data show good performance.

**Weaknesses:**

- The authors use S3OD as the name for both method and dataset. This causes confusion in the paper reading.
- The novelty of the proposed S3OD method is incremental. All components of S3OD exist in literature.
- The new S3OD dataset is AI-generated. However, all existing SOD datasets used ground truth from humans. The ground truth of S3OD dataset should come from humans.
- The experiments are unfair. All baselines were not trained on the S3OD dataset. More than 139,000+ samples of S3OD should benefit baselines.
- The authors should carefully proofread the paper. They have not described Figure 1, Figure 4 and Table 1 in the text.

**Questions:**

Please see the above comments.

**Details Of Ethics Concerns:**

N.A.

---

> ### Author Response · Authors · 2025-11-21
> **Official Comment by Authors**
>
> Thank you for your review. We appreciate your concerns, have conducted additional experiments to address them and
> provide detailed responses below.
>
> > **The authors use S3OD as the name for both method and dataset. This causes confusion in the paper reading.**
>
> We agree that this might be confusing. For better clarity we have renamed the model to S3ODNet in the paper.
>
> > **The novelty of the proposed S3OD method is incremental. All components of S3OD exist in literature.**
>
> We disagree with the reviewer that a paper's novelty only depends on new technical components/network architectures. The novelty of this paper lies in
> 1) Demonstrating a data generation pipeline that can improve tasks with limited annotated data, especially for generalization.
> 2) Introducing an ambiguity-aware model (S3ODNet) to the SOD task, which has not been considered before.
> 3) Releasing a large-scale novel, synthetic dataset for this task to allow for future improvements, now that large-scale training is a possibility.
>
> > **The new S3OD dataset is AI-generated. However, all existing SOD datasets used ground truth from humans. The ground truth of S3OD dataset should come from humans.**
>
> We disagree with this assessment.
> One of the main points of the paper is to show the usefulness of generated datasets! The reason existing datasets for SOD are small is the effort required to annotate them. DIS-5K reports 30 minutes to several hours per image for pixel-precise labels, making scaling prohibitively expensive. Even with an average label time of 10 minutes per sample, manual labeling of 139,981 images will take 23,330 hours of human labor. In contrast, our generation pipeline does not require human in the loop and can be easily scaled.
>
> Due to its synthetic nature, S3OD is a **training dataset**. All our evaluations are performed on human-annotated datasets, and we deliberately provide no test split for S3OD.
>
> The key insight is that synthetic supervision, when properly generated, produces models that generalize better than those trained on limited real data. Table 2 provides direct evidence: models trained solely on synthetic S3OD generalize better to human-annotated benchmarks compared to models trained on real data. The superior generalization to real data validates our synthetic ground truth quality and demonstrates that scale through synthesis overcomes the bottleneck of manual annotation.
>
> > **The experiments are unfair. All baselines were not trained on the S3OD dataset. More than 139,000+ samples of S3OD should benefit baselines.**
>
> We disagree. Our paper introduces a dataset (S3OD) and a method (S3ODNet). Thus, we need to perform two types of experiments to confirm the value of both:
> 1) we fix the model and vary the dataset to show that the data improves the performance.
> 2) we fix the dataset and vary the model to show that our model provides benefits independent of the data.
>
> Both types of experiments can be found in Table 2 in the paper. 1) When we train our model on the same datasets as prior work, we demonstrate better performance. 2) when we train our model on our dataset we further improve the performance, which shows that our dataset adds value.
>
> For rebuttal we go one step further: we also train prior work on our dataset. We retrained BiReftNet and MVANet on S3OD. Results are reported in Table 9 and are consistent with our main evaluation: Training on S3OD improves the generalization of all models and S3ODNet still outperforms other competitors trained in the same setup.
>
> > **The authors should carefully proofread the paper. They have not described Figure 1, Figure 4 and Table 1 in the text.**
>
> Thanks for highlighting this, fixed in the paper!

---

> > ### Comment · Reviewer_VcaT · 2025-11-28
> >
> > I thank the authors to provide the rebuttal. However, my concerns remain largely unresolved.
> >
> > In the paper, the authors introduce a synthetic dataset (S3OD dataset with 139,861 images and 1,676 unique objects). This helps boost the performance of salient object detection. This is not new. There are many works in literature which used synthetic data to improve the performance. For example, Maskfactory (Qian et al). proved that for the task of Dichotomous Image Segmentation.
> > Qian et al, Maskfactory: Towards high-quality synthetic data generation for dichotomous image segmentation, NeurIPS 2024.
> >
> > In Table 2, the authors trained S3ODNet on S3OD dataset. Meanwhile, all other baselines were trained on significantly smaller datasets. In addition, I am surprised that the authors did not include the results of PGNet proposed by Xie et al. (2022) which was introduced in the same paper as the UHRSD-TR dataset. That paper reported better performance compared to S3ODNet trained on DIS.
> > Xie et al., Pyramid grafting network for one-stage high resolution saliency detection, CVPR 2022.
> >
> > Regarding Table 3, the authors finetuned their model trained on their S3OD dataset on both the DIS-5K Qin et al. (2022) and a combination of SOD datasets (HR-SOD Zeng et al. (2019), UHRSOD Xie et al. (2022), DUTS-TR Wang et al. (2017)). Other baselines such as BiRefNet by Zheng et al. (2024) or InSpyreNet by Kim et al. (2022) were not pretrained on S3OD dataset. Furthermore, why did the authors not include more recent baselines?
> >
> > In Table 4, the authors show that S3ODNet trained on DIS + S3OD outperforms training on DIS + MaskFactory. S3OD has 139,861 images with 1,676 unique objects which is much larger than MaskFactory (10,000 images). In the MaskFactory paper, Qian et al. also showed that increasing the number of generated images leads to better performance.

---

> > > ### Author Response · Authors · 2025-12-03
> > >
> > > Thanks for the feedback. We address the main concerns below:
> > >
> > > > **In the paper, the authors introduce a synthetic dataset (S3OD dataset with 139,861 images and 1,676 unique objects). This helps boost the performance of salient object detection. This is not new. There are many works in literature which used synthetic data to improve the performance. For example, Maskfactory proved that for the task of Dichotomous Image Segmentation.**
> > >
> > > We discuss MaskFactory and other works on synthetic data generation in the related work section (lines 114-121). To the best of our knowledge, we have discussed all relevant work; however, we are open to suggestions from the reviewer.
> > >
> > > We report direct comparisons with MaskFactory in Tables 4, 10, and Figure 11. We note that MaskFactory is fundamentally different from ours: it creates variations of ground truth masks and conditions the diffusion process on augmented masks. Table 4 and Section 5.4 show that this has a fundamental limitation: it cannot expand the original data distribution and results in overfitting even more to the original dataset. On most benchmarks, the model trained with this data generalizes worse than the baseline.
> > >
> > > To summarize, the main innovations of our generation pipeline are: 1) we are first to efficiently combine generative and discriminative features; 2) we introduce iterative generation that ensures progressive mining of hard samples instead of generating all data at once without getting feedback from the model.
> > >
> > > > **In Table 2, the authors trained S3ODNet on S3OD dataset. Meanwhile, all other baselines were trained on significantly smaller datasets.**
> > >
> > > Yes. This experiment shows that models trained on small scale real data do not generalize well and scaling manual annotations is infeasible. This evaluates the efficacy of the dataset. We also report results with a fixed dataset across methods in Table 9.
> > >
> > > > **I am surprised that the authors did not include the results of PGNet proposed by Xie et al. (2022) which was introduced in the same paper as the UHRSD-TR dataset. That paper reported better performance compared to S3ODNet trained on DIS.**
> > >
> > > Unsurprisingly, PGNet trained on SOD and evaluated on SOD performs better than S3ODNet trained on DIS evaluated on SOD. This is an apples to oranges comparison, as pointed out by the reviewer themselves in a comment above.
> > >
> > > In a fair setup, where S3ODNet and PGNet are trained and evaluated on SOD our model performs better. We have added more results to Table 14.
> > >
> > > >  **The authors finetuned their model trained on their S3OD dataset on both the DIS-5K Qin et al. (2022) and a combination of SOD datasets (HR-SOD Zeng et al. (2019), UHRSOD Xie et al. (2022), DUTS-TR Wang et al. (2017)). Other baselines such as BiRefNet by Zheng et al. (2024) or InSpyreNet by Kim et al. (2022) were not pretrained on S3OD dataset.**
> > >
> > > Our main insight from the paper is that the current state of the art in saliency estimation works very well on individual datasets but does not generalize well. Thus, we purposely put the main emphasis in the evaluation on cross-dataset generalization.
> > >
> > > We emphasize that measuring overfitting on already saturated benchmarks does not provide additional insights. Nonetheless, we also provide results of finetuned models to show that this still can reach SotA performance when finetuned on real data (Lines 398-407).
> > >
> > > Table 2 and Table 10 show that S3ODNet trained on the same datasets as baselines generalizes better, and training on S3OD further improves generalization. Table 9 demonstrates that retraining other baselines improves their generalization, yet S3ODNet still generalizes better. Tables 3, 13, and 14 show the impact of finetuning on real data. Overall, this allows us to measure the dataset and model impact separately instead of only measuring performance on saturated benchmarks.
> > >
> > > > **Furthermore, why did the authors not include more recent baselines?**
> > >
> > > We are not aware of any more recent works than the ones included in the paper (typically from 2024 and 2025). Our main comparison is with BiRefNet, a widely used 2024 paper, which achieves state-of-the-art performance in SOD, DIS and COD.
> > > Since the reviewer did not provide references, we added a comparison with DIS-SAM (2025), which is a multi-stage pipeline finetuned from two larger models in Table 13.
> > >
> > > > **In Table 4, the authors show that S3ODNet trained on DIS + S3OD outperforms training on DIS + MaskFactory. S3OD has 139,861 images with 1,676 unique objects, which is much larger than MaskFactory (10,000 images).**
> > >
> > > This is not true. This experiment is fair. Lines 424-427: *"To ensure fair comparison, we train our model on DIS-5K and a mix of DIS-5K and three synthetic datasets. Since the other two synthetic datasets contain only 10,000 train images, we also subsample a subset from S3OD of the same size from the 2nd iteration of data generation."*.
> > > Even with an equal number of samples, models trained with our dataset perform better.

---

> ### Author Response · Authors · 2025-11-21
> **Baselines Table**
>
> | Method | Data | DAVIS-S | | | | HRSOD-TE | | | | UHRSOD-TE | | | | DUTS-TE | | | | DUT-OMRON | | | |
> |--------|------|---------|---------|---------|---------|----------|----------|----------|----------|-----------|----------|----------|----------|---------|---------|---------|---------|-----------|---------|---------|---------|
> | | | Fm↑ | Sα↑ | Em↑ | MAE↓ | Fm↑ | Sα↑ | Em↑ | MAE↓ | Fm↑ | Sα↑ | Em↑ | MAE↓ | Fm↑ | Sα↑ | Em↑ | MAE↓ | F_m↑ | S_α↑ | E^Φ_M↑ | MAE↓ |
> | MVANet | DIS | .907 | .929 | .959 | .016 | .902 | .919 | .930 | .033 | .922 | .926 | .941 | .032 | .852 | .877 | .893 | .042 | .711 | **.792** | .838 | .072 |
> | MVANet | S3OD | **.951** | **.958** | **.975** | **.008** | **.950** | **.948** | **.954** | **.019** | **.951** | **.943** | **.942** | **.024** | **.875** | **.893** | **.901** | **.039** | **.776** | .791 | **.873** | **.064** |
> | | | | | | | | | | | | | | | | | | | | | | |
> | BiRefNet | DIS | .919 | .936 | .961 | .014 | .887 | .915 | .926 | .031 | .922 | .924 | .937 | .032 | .860 | .886 | .910 | .036 | .744 | .819 | .835 | .054 |
> | BiRefNet | S3OD | **.963** | **.958** | **.978** | **.009** | **.956** | **.951** | **.965** | **.019** | **.955** | **.949** | **.962** | **.022** | **.928** | **.931** | **.951** | **.024** | **.845** | **.882** | **.899** | **.045** |
> | | | | | | | | | | | | | | | | | | | | | | |
> | S3ODNet| DIS | .951 | .950 | .973 | .010 | .923 | .913 | .932 | .030 | .946 | .927 | .947 | .029 | .902 | .901 | .926 | .035 | .808 | .830 | .858 | .061 |
> | S3ODNet| S3OD | **.970** | **.967** | **.988** | **.005** | **.954** | **.955** | **.972** | **.016** | **.954** | **.944** | **.961** | **.023** | **.937** | **.938** | **.962** | **.020** | **.860** | **.887** | **.911** | **.040** |

---

### Official Review · Reviewer_BL7i · 2025-11-01

**Soundness:** 3
**Presentation:** 3
**Contribution:** 3
**Rating:** 6
**Confidence:** 4

**Summary:**

S3OD targets the long-standing generalization limits in salient object detection by unifying DIS and HR-SOD through synthetic data and an ambiguity-aware model. The authors build a 139k+ high-resolution dataset via a multi-modal diffusion pipeline that extracts masks from diffusion and DINO-v3 features, then iteratively steers generation toward hard cases based on model feedback.  They propose a streamlined multi-mask decoder that explicitly models multiple valid saliency interpretations, addressing label ambiguity while keeping the architecture simple.  Trained only on synthetic data, their method improves cross-dataset generalization with 20–50% error reductions; with brief fine-tuning on real data, it reaches or surpasses state of the art across DIS and HR-SOD benchmarks.

**Strengths:**

- Multi-Modal Dataset Diffusion Pipeline that fuses diffusion feature maps, concept attention maps, and DINO-v3 representations to jointly generate images and masks, ensuring strong image–label alignment and enabling a 139k+ high-resolution synthetic set that boosts generalization.

- Ambiguity-aware architecture with a streamlined multi-mask decoder that explicitly models multiple valid interpretations.

- Iterative generation framework that is feedback-driven to prioritize challenging categories, delivering consistent gains and cross-dataset generalization, culminating in SOTA across DIS and HR-SOD and large error-rate reductions versus prior methods.

**Weaknesses:**

- The author should clarify whether mask extraction in the Multi-Modal Dataset Diffusion pipeline requires any training or calibration. If not, provide rigorous evidence of mask fidelity.

- The proposed data-generation paradigm appears tailored to binary/saliency segmentation; please evaluate transfer to camouflaged object detection (COD) or non-salient classes and report the zero-shot and fine-tuned results.

- The author should strengthen the annotation rationale with interpretable visualizations. Provide side-by-side maps of DINO-v3 features, DiT feature activations, and the resulting masks across easy and challenging scenes, plus quantitative correlations, to demonstrate complementary cues.


- On page 4, line 197, the last two terms in the loss function formula should be grouped together.

- It is recommended not to place Tables 7 and 8 at the very bottom of the page.

- Citations should be enclosed in parentheses.

- On page 2, line 059, “remains” → “remain”.

- On page 2, line 067, “features” → “feature”.

- On page.6, line.270: “order of magnitude” → “an order of magnitude”/“orders of magnitude” contain” → “contains”.

- On page.8, line.401: “siginificantly” → “significantly”.

- On page.8, line.407: “S3OD achieve” → “S3OD achieves”.

**Questions:**

- Could a teacher model be used to annotate the FLUX-generated images, then train S3OD on these teacher-labeled data, and compare its performance with the S3OD proposed in this paper?

---

> ### Author Response · Authors · 2025-11-21
> **Official Comment by Authors**
>
> Thank you for the thoughtful and positive review of our work! Below we address main questions and concerns:
>
> > **The author should clarify whether mask extraction in the Multi-Modal Dataset Diffusion pipeline requires any training or calibration. If not, provide rigorous evidence of mask fidelity.**
>
> Thanks for highlighting this. Yes, the mask extraction pipeline requires supervision. The fusion module and mask decoder are supervised using DUTS-TR, HRSOD-TR, UHRSOD-TR, and DIS-5K train sets. We then use this trained pipeline to annotate the 139K synthetic samples in S3OD.
> This is precisely why DUT-OMRON (Table 3) serves as our strongest validation dataset: it contains 5,168 images from categories completely unseen by all methods. This demonstrates our mask extraction generalizes well beyond its training distribution. We have clarified this training procedure in the revised manuscript.
>
> > **The proposed data-generation paradigm appears tailored to binary/saliency segmentation; please evaluate transfer to camouflaged object detection (COD) or non-salient classes and report the zero-shot and fine-tuned results.**
>
> We thank the reviewer for this suggestion. It turns out that our model generalizes very well to this task. We evaluated zero-shot transfer to Camouflaged Object Detection (COD). We added Table 10 (also below) and Figure 5 to the Appendix. The results show that our model trained solely on S3OD generalizes remarkably well to COD benchmarks, outperforming models trained on other datasets. When fine-tuned on COD datasets, our model achieves state-of-the-art performance on COD10K (.911 vs .888) and NC4K (.923 vs .909). Similar to SOD evaluation, we observe that the smallest benchmark, the other models are also trained on (CAMO with only 250 test images), shows saturation due to overfitting. This validates that our synthetic data teaches generalizable segmentation principles beyond salient object detection.
>
> **Table: Quantitative comparison on COD benchmarks**
>
> | Method | Data | COD10K | | | | CAMO | | | | NC4K | | | |
> |--------|------|--------|--------|--------|--------|--------|--------|--------|--------|--------|--------|--------|--------|
> | | | Fm↑ | Sα↑ | Em↑ | MAE↓ | Fm↑ | Sα↑ | Em↑ | MAE↓ | Fm↑ | Sα↑ | Em↑ | MAE↓ |
> | S3ODNet| SOD | .850 | .862 | .911 | .034 | .858 | .848 | .893 | .061 | .896 | .889 | .929 | .034 |
> | S3ODNet| DIS | .832 | .853 | .896 | .035 | .845 | .846 | .892 | .058 | .885 | .882 | .922 | .035 |
> | S3ODNet| MaskFactory | .809 | .828 | .884 | .035 | .849 | .838 | .889 | .060 | .872 | .864 | .909 | .038 |
> | S3ODNet| S3OD | **.854** | **.880** | **.920** | **.033** | **.859** | **.864** | **.906** | **.056** | **.897** | **.901** | **.936** | **.032** |
> | **Comparison with SOTA** | | | | | | | | | | | | | |
> | FSPNet | COD | .769 | .851 | .895 | .026 | .830 | .856 | .899 | .050 | .843 | .879 | .915 | .035 |
> | BiRefNet | COD | .888 | .913 | .960 | .014 | .904 | **.904** | **.954** | **.030** | .909 | .914 | .953 | .023 |
> | S3ODNet| S3OD+COD | **.911** | **.923** | **.970** | **.012** | **.908** | .903 | .949 | .031 | **.923** | **.920** | **.961** | **.020** |
>
> > **The author should strengthen the annotation rationale with interpretable visualizations. Provide side-by-side maps of DINO-v3 features, DiT feature activations, and the resulting masks across easy and challenging scenes, plus quantitative correlations, to demonstrate complementary cues.**
>
> We agree this would clarify the choice of the feature sources and have added Figure 6 to the revised manuscript showing side-by-side visualizations of each modality across diverse scenes. Concept attention maps provide coarse but strong foreground/background separation based on semantic tokens. DINO-v3 features encode fine-grained semantic similarity (nearby regions with similar semantics cluster together), and being trainable, adapt to task-specific requirements. FLUX DiT features capture spatial scene parsing information encoded during image generation, including object boundaries, semantic regions, and background structure (note: high-dimensional features are visualized via PCA, which partially reduces interpretability).
>
> > **Could a teacher model be used to annotate the FLUX-generated images, then train S3OD on these teacher-labeled data, and compare its performance with the S3OD proposed in this paper?**
>
> The current pipeline can be viewed as a variant of a teacher model that also incorporates additional information for the generation process. We would like to clarify if the reviewer means training S3ODNet on real data and using it to pseudolabel the generated images? If the reviewer thinks it can help to improve the method evaluation, we can add this experiment.
>
> > **Minor mistakes in the text**
>
> Thanks for highlighting this! We have fixed these in the updated manuscript

---

> > ### Comment · Reviewer_BL7i · 2025-11-28
> > **Official Comment**
> >
> > Thank you for addressing my concerns. I am satisfied with the revisions made to the paper and have decided to maintain my positive rating.

---

### Official Review · Reviewer_7iqD · 2025-11-02

**Soundness:** 3
**Presentation:** 3
**Contribution:** 3
**Rating:** 6
**Confidence:** 5

**Summary:**

This paper introduces S3OD, a new synthetic data generation pipeline and ambiguity-aware model for salient object detection (SOD). The approach leverages multi-modal diffusion models to generate over 139,000 high-quality, high-resolution images and corresponding masks that capture diverse, ambiguous segmentation scenarios. S3OD combines iterative, feedback-driven data sampling and a streamlined multi-mask architecture built on a DINO-v3 backbone, aiming to improve generalization across both dichotomous segmentation (DIS) and high-resolution SOD (HR-SOD) tasks. Empirical evaluation demonstrates strong cross-dataset transfer and new state-of-the-art results on multiple SOD/DIS benchmarks. The dataset and code will be released.

**Strengths:**

**1) Scale and Diversity of Synthetic Data (Figure 1, Table 1, Figure 4, Figure 6):** S3OD delivers an order-of-magnitude increase in dataset scale for SOD, with 139k+ images spanning 1676 unique objects and a wide spectrum of scene types, lighting, and occlusions, as seen qualitatively in Figures 1, 4, and 6 and quantitatively in Table 1. Manually verified mask quality and data curation strategies, including filtering with VLMs, result in a synthetic dataset that rivals or exceeds real sets in annotation fidelity.

**2) Innovative Data Generation Pipeline (Figure 3):** The multi-modal pipeline fuses diffusion latent features, concept attention maps, and DINO-v3 features to create dense, pixel-precise masks without teacher bottlenecks or reliance on pre-existing mask libraries. The iterative feedback loop dynamically adjusts category sampling based on model performance (see Figure 3 and Section 4.2), which is a notable step forward in dataset synthesis methodology.

**3) Ambiguity-Aware Model Architecture (Figure 2):** The proposed multi-mask head on top of a DINO-v3/DPT backbone enables the network to represent inherent annotation ambiguity—a factor often overlooked in SOD. Figure 2 provides a clear and helpful visualization of the model’s architecture, showing multiple candidate mask outputs and predicted IoU for selection at inference.

**Weaknesses:**

**Potential for Domain Overfitting or Synthetic-“Leakage” Not Fully Addressed:**

1. Although the cross-dataset generalization is well documented, concerns about overfitting to synthetic artifacts (such as those possibly present in highly artificial or LLM-generated prompts) are only partly mitigated by filtering and photo-realism tuning (see Figure 7 and Section B). There is no explicit domain gap or bias quantification (such as t-SNE/UMAP distributions, or model calibration metrics) to back up claims about synthetic-to-real transfer.

2. Table 9 claims strong Inception/FID scores, but these are image-level metrics—less relevant than mask consistency or class-level error for segmentation evaluation.

**Limited Theoretical Justification for Multi-Modal Fusion and Mask Decoding (Section 3.1, Equations):**

1. While practical motivation is given, the paper glosses over the theoretical rationale for fusing FLUX DiT, concept attention, and DINO-v3 features in the supervision pipeline. The process is described at a systems level (“projected to a common 256-dimensional space via separate convolutional branches...”) but lacks rigorous analysis of why this fusion is optimal, how information is disentangled, or how fusion improves over single-modality supervision. Further, the formulation of loss weighting and decay (e.g., $\lambda_{\text{mask}}=10$, $\gamma=0.2$) is heuristically chosen; no ablations or theoretical discussion support these choices.

2. The mask selection (best out of N) and loss propagation scheme is borrowed from multiple-choice learning but could benefit from a more detailed analysis or ablation to determine how many branches (N) are optimal and how performance scales with it.

**Questions:**

Please see Weaknesses.

---

> ### Author Response · Authors · 2025-11-21
> **Official Comment by Authors**
>
> Thank you for the thoughtful and positive review of our work! Below we address main questions and concerns:
>
>
> > **There is no explicit domain gap or bias quantification (such as t-SNE/UMAP distributions, or model calibration metrics)**
>
> Thanks, we agree that additional domain gap metrics would further strengthen the analysis of the dataset quality. To quantitatively evaluate the synthetic-to-real domain gap, we compute the quality and coverage of the samples produced by a generative model versus a combination of the SOD and DIS datasets. We observe that both synthetic images and masks closely follow the real distribution, in contrast to other methods, which model only part of it. Further, UMAP projections of DINO-v3 image and mask features demonstrate that S3OD samples cover a larger region of the real data manifold compared to MaskFactory. The reduced domain gap directly explains the superior generalization of the models trained on S3OD. We have added the results as Figure 11 in Appendix E.
>
> > **Table 9 claims strong Inception/FID scores, but these are image-level metrics—less relevant than mask consistency or class-level error for segmentation evaluation.**
>
> We interpret 'mask consistency' as annotation quality. We report the manual verification of the quality of datasets in the paper, which shows 98.6% masks are labeled as consistent (Appendix E).
>
> > **Lacks rigorous analysis [...] how fusion improves over single-modality supervision**
>
> This is one of our main ablation experiments in the paper. Table 6 shows that each feature type contributes to the final performance. For the rebuttal we have added visual exemplars of different features sources on various data samples (Figure 6).
>
> >  **The formulation of loss weighting and decay (e.g., λ, γ) is heuristically chosen; no ablations or theoretical discussion support these choices.**
>
> A detailed discussion of these parameters can be found in Rupprecht et al. (2017). In practice the algorithm was not very sensitive to these parameters but we empirically observe that removing λreg results in one of the branches stopping to recieve the graidents, while too high λ forces the model to predict identical masks ignoring the ambiguity.
>
> > **The paper glosses over the theoretical rationale for fusing FLUX DiT, concept attention, and DINO-v3 features in the supervision pipeline**
>
> The beginning of Section 4.1 discusses the rational of these choices. \
> Each captures complementary information: (1) FLUX DiT features encode spatial layout understanding from the generative process, (2) Concept attention provides explicit semantic grounding to specific object categories, (3) DINO-v3 captures fine-grained visual semantics robust to synthetic-to-real domain shift.
> Fusion architecture: The 256-dimensional projection space is chosen to match the original DPT decoder implementation, ensuring computational parity with baseline methods (Table 12). Each modality is independently projected before concatenation to preserve their distinct feature spaces, then fused through convolutional layers.
>
> > **The mask selection (best out of N) and loss propagation scheme is borrowed from multiple-choice learning but could benefit from a more detailed analysis or ablation to determine how many branches (N) are optimal and how performance scales with it.**
>
> Table 7 provides this ablation: we evaluate N ∈ {1, 2, 3} mask predictions, showing N=3 achieves optimal performance.
> Through visual inspection of ambiguous SOD images, we observe that annotation ambiguity typically involves 2-3 plausible interpretations (e.g., unclear foreground boundaries, multiple candidate objects, complex object interactions). Cases requiring N>3 are exceptionally rare. This explains the empirical results: N=1 cannot model ambiguity, N=2 captures some but not all ambiguous cases, and N=3 provides sufficient coverage without unnecessary computational overhead.

---

> ### Comment · Reviewer_7iqD · 2025-11-21
> **All concerns are addressed**
>
> All concerns are addressed, thanks for rebuttal.

---

> > ### Author Response · Authors · 2025-11-27
> >
> > Dear Reviewer 7iqD,
> >
> > Thank you very much for your time and valuable feedback! We are glad that our updates were helpful. If you have any further questions, please don't hesitate to ask.
> >
> > Best regards,
> > Authors

---

### Author Response · Authors · 2025-12-03
**Summary of Rebuttal and Discussion Phase**

We thank the reviewers (**7iqD**, **BL7i**, **VcaT**, **n7Sk**) for insightful feedback and efforts to further improve the paper.

Our work targets the generalization across different Salient Object Detection (SOD) tasks through scaling up data volumes with synthetic data and introducing a simple yet effective model architecture that handles multiple hypotheses.
The novelty of our paper lies in:

1. Demonstrating a data generation pipeline that can improve tasks with limited annotated data, especially for generalization.
2. Introducing an ambiguity-aware model (S3ODNet) to the SOD task, which has not been considered before.
3. Releasing a large-scale novel, synthetic dataset for this task to allow for future improvements, now that large-scale training is a possibility.

Through the active discussion, we have further improved the evaluation of: 1) the dataset quality (domain gap analysis); 2) generalization (COD); 3) SOTA performance (comparison with more benchmarks), and a multi-mask decoder analysis (ablation study and theoretical justification). We believe these changes significantly improve the quality and readability of the paper. We have received a positive response from three reviewers and an active engagement from everyone, resulting in a productive discussion.

### Rebuttal Summary

The key points of discussion were:
* **Quantitative analysis of the domain gap between real and synthetic data (7iqD, n7Sk).** We further evaluate the synthetic-to-real domain gap by computing the quality and coverage of the samples produced by a generative model versus a combination of SOD and DIS datasets. We also provide UMAP visualizations of S3OD vs MaskFactory. Both reviewers explicitly mention that they are satisfied with the response (*"a compelling way to strengthen the domain gap analysis"*).

* **Clarify dataset (S3OD) vs model (S3ODNet) impact on performance (VcaT, n7Sk).** To further disentangle the contributions of the new dataset and the S3ODNet model we retrain baselines on S3OD dataset. Results are consistent with our main evaluation: Training on S3OD improves the generalization of all models and S3ODNet still outperforms other competitors trained in the same setup. Reviewer n7Sk highlights the value of additional evaluation. Reviewer VcaT had additional questions, which we addressed after the discussion was closed for reviewers.

* **Multi-Mask Decoder Analysis (7iqD, n7Sk)** We sincerely thank the reviewers for the good suggestions on highlighting the core difference between standard multi-choice learning frameworks and our decoder. We have added an ablation of regularization loss and a detailed analysis to Supplementary I.

* **Generalization Beyond SOD and DIS (BL7i)** We thank you for the suggestion and provide an extended evaluation of the generalization of S3ODNet to camouflaged object detection (COD). It turns out that our model generalizes very well to this task. The results show that our model trained solely on S3OD generalizes remarkably well to COD benchmarks, outperforming models trained on other datasets. When fine-tuned on COD datasets, our model achieves state-of-the-art performance on COD10K (.911 vs .888) and NC4K (.923 vs .909).

---

### Meta-Review · Area_Chair_sRKc · 2026-01-08

**Summary:**

This paper proposes a system-level solution to improve cross-dataset and cross-task generalization in salient object detection by combining large-scale synthetic data generation with an ambiguity-aware multi-mask decoding architecture. The authors introduce S3OD, a large synthetic dataset generated via a multi-modal diffusion-based pipeline, and demonstrate that models trained solely on this synthetic data achieve substantial generalization gains across SOD, DIS, HR-SOD, and COD benchmarks, with fine-tuned models reaching or surpassing state of the art.

The main points of discussion during review concern (i) the validity of synthetic-to-real transfer, (ii) whether performance gains are confounded by data scale or pretrained backbones, and (iii) the degree of algorithmic novelty of the ambiguity-aware decoder. After carefully reviewing the paper, rebuttal, and discussion, I conclude that the empirical evidence and additional analyses sufficiently support the paper’s central claims, and that the remaining concerns primarily relate to theoretical depth and scope rather than to soundness or experimental validity.

**Reviewer Concerns:**

Concerns Addressed

1. Synthetic-to-real domain gap and dataset quality.
Multiple reviewers raised concerns regarding whether synthetic data truly captures the real data distribution. These concerns have been convincingly addressed through additional domain gap analyses, including representation-level evaluations (e.g., UMAP visualizations of DINO-v3 features) and quality/coverage metrics comparing synthetic and real datasets. Several reviewers explicitly stated that these additions satisfactorily strengthen the justification for synthetic-to-real generalization.
2. Disentangling dataset scale from model contributions.
Reviewers questioned whether performance improvements stem mainly from larger data volume or stronger pretrained backbones. The authors addressed this through controlled experiments that fix either the dataset or the model, as well as by retraining prior methods on the proposed S3OD dataset. These results consistently show that (i) S3OD improves generalization across different architectures, and (ii) the proposed model provides gains beyond data scale alone.
3. Generality beyond standard SOD benchmarks.
Concerns about task specificity were mitigated by additional zero-shot and fine-tuned evaluations on camouflaged object detection (COD), demonstrating that the learned representations generalize beyond binary salient object detection and further supporting the broader applicability of the synthetic data generation framework.

Remaining but Non-blocking Concerns

1. Algorithmic novelty and theoretical grounding of the ambiguity-aware decoder.
While the ambiguity-aware multi-mask decoder is empirically effective, its core mechanism is closely related to existing multiple-choice learning and multi-point estimation frameworks. The additional theoretical discussion (grounded in prior work) improves clarity but does not elevate this component to a standalone theoretical contribution. Importantly, this limitation does not undermine the paper’s primary system-level contribution or empirical findings.
2. Computational cost and reproducibility.
The data generation pipeline and training procedure involve substantial computational resources. The authors now explicitly acknowledge this limitation, provide alternative hardware configurations, and commit to releasing code, models, and data. While full from-scratch reproduction may be costly, this concern affects practicality rather than validity.

**Reviewer Scores:**

- Reviewer 7iqD: Likely to maintain a marginally positive score after confirming that all major concerns were addressed.
- Reviewer BL7i: Explicitly indicated satisfaction with the revisions and would maintain or slightly strengthen a positive score.
- Reviewer n7Sk: Appears to move from borderline acceptance to a clearer weak accept, contingent on incorporating the clarified theoretical discussion and reproducibility statements.
- Reviewer VcaT: Remains unconvinced regarding novelty and the use of synthetic data; however, this position is an outlier and reflects a stricter novelty criterion that is less aligned with the paper’s system-level and data-centric contributions, especially in light of the additional controlled experiments and clarifications.

---

### Decision · Program_Chairs · 2026-01-26

Accept (Poster)